# Where Does It Exist from the Low-Altitude: Spatial Aerial Video Grounding

**Yang Zhan**    **Yuan Yuan**[*]
School of Artificial Intelligence, Optics and Electronics (iOPEN)
Northwestern Polytechnical University
zhanyangnwpu@gmail.com, y.yuan1.ieee@gmail.com

## Abstract

The task of localizing an object's spatial tube based on language instructions and video, known as spatial video grounding (SVG), has attracted widespread interest. Existing SVG tasks have focused on ego-centric fixed front perspective and simple scenes, which only involved a very limited view and environment. However, UAV-based SVG remains underexplored, which neglects the inherent disparities in drone movement and the complexity of aerial object localization. To facilitate research in this field, we introduce the novel spatial aerial video grounding (SAVG) task. Specifically, we meticulously construct a large-scale benchmark, **UAV-SVG**, which contains over 2 million frames and offers 216 highly diverse target categories. To address the disparities and challenges posed by complex aerial environments, we propose a new end-to-end transformer architecture, coined **SAVG-DETR**. The innovations are three-fold. 1) To overcome the computational explosion of self-attention when introducing multi-scale features, our encoder efficiently decouples the multi-modality and multi-scale spatio-temporal modeling into intra-scale multi-modality interaction and cross-scale visual-only fusion. 2) To enhance small object grounding ability, we propose the language modulation module to integrate multi-scale information into language features and the multi-level progressive spatial decoder to decode from high to low level. The decoding stage for the lower-level vision-language features is gradually increased. 3) To improve the prediction consistency across frames, we design the decoding paradigm based on offset generation. At each decoding stage, we utilize reference anchors to constrict the grounding region, use context-rich object queries to predict offsets, and update reference anchors for the next stage. From coarse to fine, our SAVG-DETR gradually bridges the modality gap and iteratively refines reference anchors of the referred object, eventually grounding the spatial tube. Extensive experiments demonstrate that our SAVG-DETR significantly outperforms existing state-of-the-art methods. The dataset and code will be available at *here*.

## 1 Introduction

Grounding objects with natural language in visual contexts is a fundamental and important task in multi-modal understanding [1, 2, 3]. Recently, the spatial video grounding (SVG) has drawn significant attention [4, 5]. However, the SVG task has predominantly focused on the ego-centric fixed front perspective and simple scene [6, 7, 8]. As shown in Figure 6 of the supplementary material, they only provide a very limited view and environment. This means that existing methods can only perform target localization in simple scenes on the ground. This overlooks another important application scenario: moving aerial platforms in the sky [9, 10, 11]. As the low-altitude economy

---

[*]Corresponding author.

39th Conference on Neural Information Processing Systems (NeurIPS 2025).

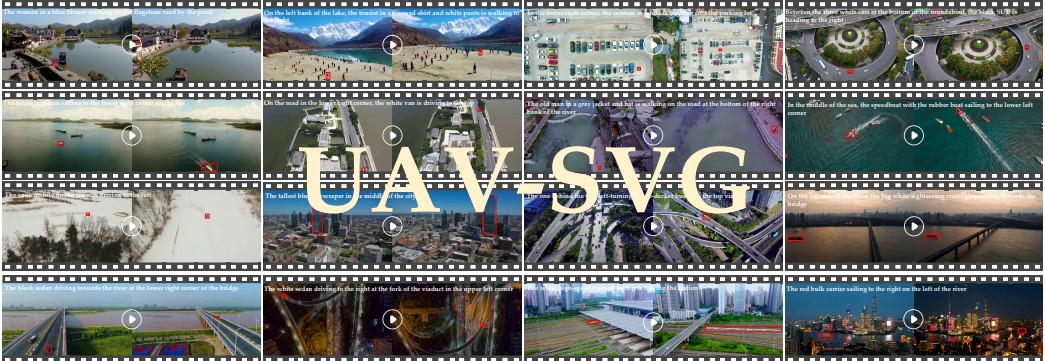

Figure 1: An overview of the UAV-SVG dataset. SAVG grounds the referred object's spatial tube in the complex aerial scene by a natural language query. UAV-SVG presents distinctive challenges, including camera motion, low resolution, illumination variations, aspect ratio variations, viewpoint changes, rotation, etc. More detailed analysis is in Sec. B.3 of the supplementary material.

takes off, many tasks currently need to be performed in the sky [12], such as UAV-based goods delivery, traffic/security patrol, and scenery tours [13, 14, 15]. Spatial aerial video grounding (SAVG) emerges as a groundbreaking task. We can use UAVs to localize specific objects in various scenarios from the sky view and obtain a much more holistic grounding.

To date, existing benchmarks (VID-sentence [6], VidSTG [7], HC-STVG [8]) have predominantly been confined to small-scale scenarios (Figure 6). The number of objects within the video is limited, and the referred object covers a significant portion of the frame image. SVG in moving aerial platforms is quite different, presenting unique technical challenges: 1) Most objects only contain a few pixels (1-200 pixels or so). The approaches in natural scenes without multi-scale feature learning cannot deal with them effectively. 2) Aerial videos usually have a wide field of view and a large scene scale, containing dense and numerous objects. The irrelevant or confusing information increases significantly. 3) Objects may be subject to a range of environmental disturbances, such as occlusion from trees, shadows cast by sunlight, and low light at night. The discriminability of the referred object diminishes significantly. 4) UAV may move rapidly, causing viewpoint changes between adjacent frames, and the object may also move quickly. It is hard to accurately and consistently ground objects. To foster our proposed SAVG task, we contribute a new sizeable benchmark, named **UAV-SVG**. UAV-SVG uses the million-scale tracking dataset [9] as the video source. To guarantee the high quality, referring expressions are generated using a combination of manual annotation and the advanced Gemini model [16]. As shown in Table 1, UAV-SVG contains over 2 million frames, 17,820 video–query pairs, and 216 highly diverse object categories. As shown in Figure 1, unlike ground-fixed or hand-held shooting sources, the outdoor aerial view encompasses vast areas.

Existing advanced end-to-end methods [17, 18, 19, 20] usually follow three steps: 1) transformer encoder for multi-modality 3D spatio-temporal modeling, 2) transformer decoder for mining information from encoded vision-language features by learnable queries, and 3) a prediction head for regressing queries to obtain the spatial tube. In the field of aerial object localization, most works [21, 22, 23] use multi-scale visual features. If we introduce multi-scale visual features into the existing encoder, the self-attention of computing multi-modality multi-scale 3D spatio-temporal features will be computationally expensive and unaffordable. Meanwhile, it is difficult to capture the object-related region details efficiently in the complete and lengthy multi-modality multi-scale sequence during the decoding stage, especially the small objects. Moreover, due to the complicated movement of UAVs, the prediction consistency across frames of the grounding model is more demanding.

To address aforementioned problems, we introduce a new end-to-end transformer architecture, termed **SAVG-DETR**. Our core consists of a multi-modality multi-scale spatio-temporal encoder for cross-modal cross-scale feature fusion and alignment, and a hierarchical progressive decoder for efficient spatial tube prediction. ***The first key design*** is that the vanilla spatio-temporal encoder is decoupled into two branches: intra-scale multi-modality interaction with a single high-level scale and cross-scale visual-only fusion with different scales. Our encoder can capture conceptual entities on high-level features with richer semantic concepts and integrate more object details from low-level features, which is convenient for the subsequent decoder to localize the object. In addition, this decoupling

Table 1: Comparison of spatial video grounding datasets, where 'Exp.' indicates the expression.

| Dataset | Videos Num. | Frame Num. | Total Duration | Object Classes | Motion Classes | Language Num. | Vocab | Exp. Length | Train/Val/Test Partition | Target or Scene | Shot & View |
|---|---|---|---|---|---|---|---|---|---|---|---|
| VID-sentence [6]ACL'19 | 7,654 | 59K | 11.1 h | 30 | ✗ | 7,654 | 1,823 | 13.18 | 86%/7%/7% | Animal & Vehicle | Fixed / Handheld & Front |
| VidSTG [7]CVPR'20 | 6,924 | 7.1M | 69.1 h | 79 | ✗ | 99,943 | 1,881 | 10.12 | 80%/10%/10% | Human & Animal | Fixed / Handheld & Front |
| HC-STVG [8]TCSVT'21 | 5,660 | 3M | 31.4 h | 1 | ✗ | 5,660 | 2,289 | 17.27 | 80%/0%/20% | Human | Movie Clips |
| **UAV-SVG** | **3,564** | **2M** | **18.7 h** | **216** | **73** | **17,820** | **3,243** | **16.39** | **79%/5%/16%** | **Wild** | **Moving Aerial & Bird's-Eye** |

strategy avoids an explosion in the computation of multi-modality multi-scale 3D spatio-temporal features. ***The second key design*** is multi-level progressive spatial decoder, which decodes from high to low level and gradually increases the number of decoding layers for lower-level features. We devise the multi-level language modulation module to integrate multi-scale information into language features. The spatial decoder utilizes multi-level language-vision features to guide queries to decode more relevant spatial information. ***The third key design*** is the decoding paradigm based on offset generation. Unlike existing methods, we utilize reference anchors as positional embedding to constrict the grounding region, use queries to predict offsets, and update reference anchors at each decoding stage. We design the query and position generator to yield context-rich object queries and initial reference anchor boxes. This paradigm improves the consistency of the prediction. Furthermore, we adopt larger auxiliary bounding boxes to calculate losses, which is more effective for small objects. To demonstrate the effectiveness of our approach, we conduct comprehensive ablation studies and benchmark many state-of-the-art methods.

***Contributions: (i)*** We highlight the significance of deploying spatial video grounding in aerial scenes and introduce a challenging benchmark, UAV-SVG, characterized by unique properties and challenges that set it apart from existing datasets. ***(ii)*** To overcome the computational explosion, our encoder efficiently decouples the multi-modality and multi-scale spatio-temporal modeling into intra-scale multi-modality interaction and cross-scale visual-only fusion. ***(iii)*** To enhance small object grounding, we propose the modulation module to integrate multi-scale information into language features and the multi-level progressive spatial decoder to decode from high level to low level. ***(iv)*** To improve the prediction consistency, we design the decoding paradigm based on offset generation. At each decoding stage, we use context-rich queries to predict offsets and update reference anchors for the next stage. ***(v)*** Extensive experiments show that our method significantly outperforms all baselines. Comprehensive ablation studies and detailed analyses provide new ideas and useful insights.

## 2   Related work

**Image-Based Visual Grounding in Aerial.**   The visual grounding in aerial [24, 25, 26] is mainly focused on the satellite remote sensing scenario, such as MGVLF [27], QAMFN [28], LPVA [29], RMSIN [30]. In aerial scenes, images with a large field of view and complex spatial scales are often encountered. Existing methods propose a multi-granularity visual language fusion module [27], a multi-level feature enhancement decoder [29], a multi-scale cross-modal alignment module [26], or a rotated multi-scale interaction network [30] to capture remote sensing vision-language features and achieve improved performance effectively.

**Video-Based Referring Expression Comprehension.**   This task aims to detect the unique object or region in each video frame using a phrase or expression that describes the target attribute. The earlier tracking-based approaches generate phrase-relevant region proposals [31, 32] and transform the visual tracking framework into natural language tracking [33, 34, 32]. Detection-based methods [35, 36, 37, 38, 39] do not rely on visual region proposals and directly localize objects in each frame. Recent one-stage frameworks, like Co-Grounding [35], DCNet [36], ConFormer [38], and MILCGF-Net [39], have focused on temporal correlation, inter-frame correlation, fine-grained patch-word alignment, phrase-region alignment, and image-language inter-modality dense associations.

**Video Object Grounding.**   This task aims to localize all objects in the video referred to in the natural language query. The number of target boxes output in each frame may vary and is not limited to only one. The VOGNet framework [40] adopts self-attention with relative position encoding to model object relations. Subsequently, the weakly supervised video object grounding (WSVOG) [41] introduces context-aware object stabilizer module and cross-modal alignment knowledge transfer modules to achieve stable context learning. The UMA framework [42] considers rich contextual

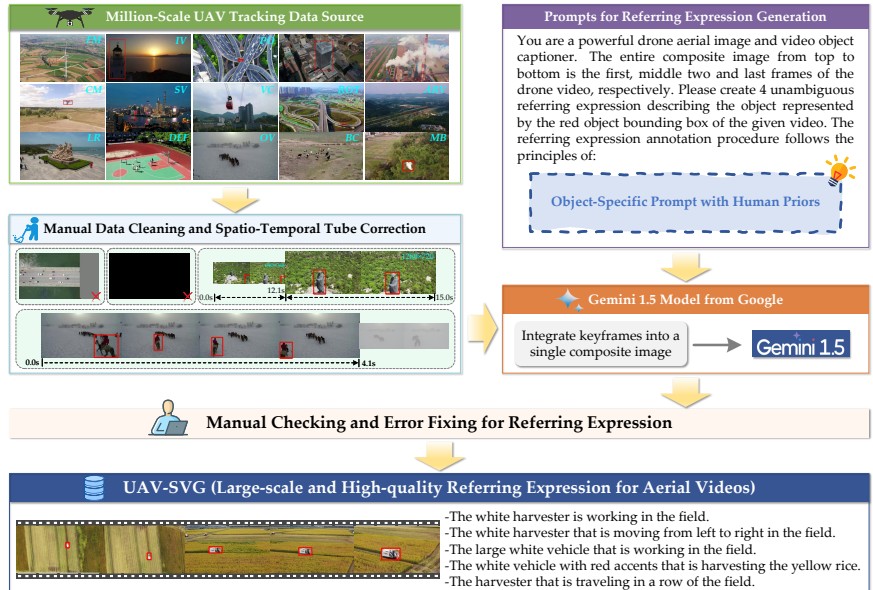

Figure 2: The construction pipeline of our UAV-SVG benchmark: Step 1) manual data cleaning and spatio-temporal tube correction, Step 2) referring expression generation, and Step 3) manual checking and error fixing. OV indicates that the object is out of view, and the other 12 characteristics are detailed in Sec.B.3 of the supplementary material.

information of the same object to explicitly learn textual and visual uni-modal associations. The unified causal framework [4] proposes spatial-temporal adversarial contrastive learning and backdoor adjustment for causal intervention to learn object-relevant association.

**Spatio-Temporal Video Grounding.** This task aims to localize the spatio-temporal tube of the unique referred object or region in the untrimmed video by a sentence query. The earlier two-stage methods [7, 8, 43] employ a pre-trained detector to generate candidate region proposals. Inspired by the DETR model [44], recent works [17, 20] use the one-stage paradigm, which does not depend on the quality of the pre-trained detector. TubeDETR [17] and STCAT [18] design transformer-based video-text encoders and a space-time decoder for joint modeling of spatio-temporal and multi-modal interactions. To solve the heavy computational complexity and insufficient spatio-temporal interactions, SGFDN [45] decomposes 3D spatio-temporal features into 2D motion and 1D object embedding. A cross-stream collaborative reasoning framework [46] decomposes static and dynamic vision-language flows to capture object appearance and motion cues, respectively. CG-STVG [19] contains instance context generation and refinement modules to capture instance visual context information and eliminate irrelevant or harmful information respectively. VideoGrounding-DINO [20] utilizes pre-trained Grounding DINO to achieve powerful open-vocabulary performance.

## 3 UAV-SVG Benchmark

To the best of our knowledge, only aerial videos with natural language annotations are CapERA [47] and WebUAV-3M [9]. However, CapERA's annotations are video-level captions and cannot refer to objects or regions. WebUAV-3M is a single-object tracking dataset with spatial tubes and language descriptions of first-frame objects. However, this dataset cannot be directly and perfectly adapted to the SAVG task due to many hard defects. To this end, we contribute a new dataset by annotating video objects in the WebUAV-3M with the latest Gemini model [16], namely UAV-SVG.

**Dataset Annotation.** The construction pipeline of the newly proposed UAV-SVG is shown in Figure 2. We choose WebUAV-3M as our data source for two main reasons. First, it is the largest public UAV tracking dataset to date, containing videos with complex scenes and diverse categories. Second, object descriptions of the first frame and bounding boxes of the target sequence are provided to avoid labor-intensive annotation for spatial and textual labels of the dataset. The detailed process are provided in Sec. B.1 of the supplementary material.

**Analyses of UAV-SVG.** This section analyzes the salient differences between UAV-SVG and existing video grounding benchmarks, including video resolution, scales of bounding boxes, lengths of expressions, quantities of object classes, target position distribution, and data statistics. We provide detailed analyses in Sec. B.2 of the supplementary material. UAV-SVG contains over 2.01 million frames across 3,564 videos and offers 216 highly diverse object categories. The total duration and average duration of videos are 18.7h and 18.86s, as shown in Table 1. There are 17,820 video-sentence-tube triples in our constructed UAV-SVG dataset. The split of training, validation, and testing is shown in Table 6 of the supplementary material. Spatial video grounding in natural scenes is quite different from that in aerial scenes. Specifically, 12 characteristics and challenges are detailed in Sec. B.3 of the supplementary material. We believe that unique characteristics of UAV-SVG can open the door for the SAVG with practically useful and broader real-life applications.

## 4 Methodology

The framework of our SAVG-DETR is shown in Figure 3. Problem definition is shown in Sec.4.1. We first extract multi-scale video and language features from pre-trained backbones (Sec. 4.2). Different from the original DETR [44], our encoder (Sec. 4.3) decouples the multi-modality multi-scale spatio-temporal modeling into intra-scale multi-modality interaction and cross-scale visual-only fusion. Learnable video tokens and frame tokens during multi-modality interaction are used to generate initial object queries and position embeddings for the decoder. Subsequently, our decoder (Sec. 4.4) utilizes fused multi-scale features to modulate language features and progressively decode multi-level language-video features from high to low level. In the decoder, we use object queries to generate position offsets to constantly update the object spatial tube. Finally, we introduce a scaling factor to generate larger auxiliary bounding boxes and improve spatial grounding loss function (Sec. C.3 of the supplementary material).

### 4.1 Problem Definition

The spatial aerial video grounding task aims to localize the referred object sequence in an aerial video by integrating vision-language information. In contrast to spatial grounding, which focuses on localizing objects in a single frame image, this task extends the temporal dimension on this concept. This means understanding where the objects are in each frame and how they move over time from the aerial view. Given an aerial video $V \in \mathbb{R}^{T \times C \times H \times W}$ with $T$ consecutive frames, $C$ channels, and $H \times W$ spatial resolution, respectively, and a natural language description $S$ depicting one object existing in $V$. The SAVG problem can be defined as localizing a spatial tube $\boldsymbol{B} = \{\boldsymbol{b}_t\}_{t=1}^T$ of the specific object referred to by the description $S$, where $\boldsymbol{b}_t = (x_t, y_t, w_t, h_t)$ represents a bounding box in the $t$-th frame. $(x_t, y_t)$ are the coordinates of the center and $(w_t, h_t)$ are the width and height of the bounding box.

### 4.2 Aerial Video-Text Feature Extractor

Following the existing literature [17], we use ResNet as the visual backbone to extract the aerial visual features for each frame. The visual encoder is initialized with weights from MDETR [48] pre-trained on Flickr30k [49], MS COCO [50], and Visual Genome [51]. We use multi-scale visual feature maps $\{\boldsymbol{S}_3, \boldsymbol{S}_4, \boldsymbol{S}_5\}$ extracted from the last three stages of the backbone as the input of the multi-modality multi-scale spatio-temporal encoder. Formally, we flatten multi-scale visual feature maps $\boldsymbol{S}_i$ from $\mathbb{R}^{c_i \times h_i \times w_i}$ to $\mathbb{R}^{c_i \times h_i w_i}$. Three $1 \times 1$ convolutional layers are used to project them into the same channel dimension $C = 256$, thus we get multi-scale visual features $\boldsymbol{F}_i \in \mathbb{R}^{C \times N_{vi}} (N_{vi} = h_i w_i)$. We take as input a set of multi-scale aerial video features $\boldsymbol{F}_{vi} \in \mathbb{R}^{T \times C \times N_{vi}} (i = 3, 4, 5)$ from the visual encoder for all $T$ frames of the input aerial video.

For the language encoder, we leverage the pre-trained RoBERTa [52] to convert the natural language description into an output with $N_l$ tokens and $C_l$ channel dimension. One linear layer is used to project it into the channel dimension $C$, thus we can get language features $\boldsymbol{F}_l \in \mathbb{R}^{C \times N_l}$.

### 4.3 Multi-Modality Multi-Scale Spatio-Temporal Encoder

Inspired by [53, 54], we introduce multi-scale visual features into the traditional encoder. However, computing the self-attention between multi-scale visual features and language features for each frame

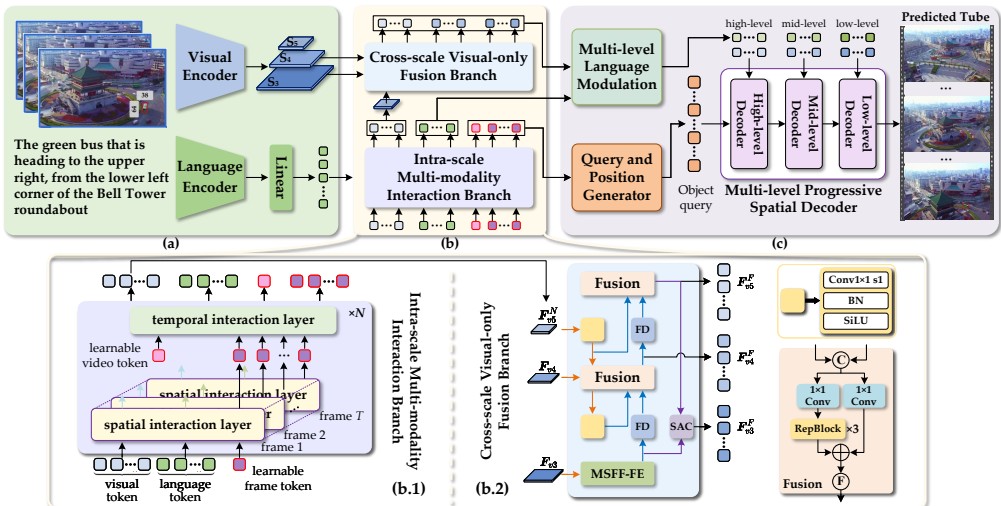

Figure 3: The framework of SAVG-DETR consists of (a) aerial video-text feature extractor, (b) multi-modality multi-scale spatio-temporal encoder, and (c) hierarchical progressive decoder.

is computationally expensive and unaffordable. To overcome this problem, we decouple the encoder into intra-scale multi-modality interaction and cross-scale visual-only fusion. In order to intuitively understand, we show the detailed design idea in Sec. C.1 of the supplementary material.

**Intra-scale Multi-modality Interaction Branch.** This branch aims to model multi-modality interactions between the language features $\boldsymbol{F}_l$ and high-level aerial video features $\boldsymbol{F}_{v5} = \left\{ \boldsymbol{F}_{v5t} \in \mathbb{R}^{C \times N_{v5}} \right\}_{t=1}^{T}$. Specifically, the branch consists of a $N$ layer encoder. Each layer starts with a spatial interaction layer followed by a temporal interaction layer, as shown in Figure 3 (b.1). We introduce a learnable embedding $\boldsymbol{F}_t^f \in \mathbb{R}^{C \times 1}$ (namely frame token) in $t$-th frame to capture the spatial context of the referred object through intra-modality and inter-modality interactions. Frame tokens $\boldsymbol{F}^f = \left\{ \boldsymbol{F}_t^f \right\}_{t=1}^{T}$ fuse information across spatial dimensions of visual and textual modalities. The spatial interaction layer conducts local spatial modeling for each frame but lacks global temporal modeling. The temporal interaction layer applies the self-attention across temporal dimensions between frames tokens. Similarly, we introduce a learnable embedding $\boldsymbol{F}^v \in \mathbb{R}^{1 \times C}$ (namely video token) to capture the global aerial video-text context.

**Cross-scale Visual-only Fusion Branch.** The objective of this branch is to fully extract both high-level semantic information and detailed object information from multi-scale visual features for efficient localization of the grounded object. First, it transfers the rich contextual semantics embedded in high-level visual features after multimodal interaction to enhance low-level features. Moreover, it injects the object localization details contained in low-level features into high-level features via fusion. To address the challenges of extensive small objects and avoid a large computational cost generated by this branch, we derive solutions from the popular real-time detection transformer [21, 22, 55, 23]. This branch mainly includes the two-scale fusion module, multi-scale feature fusion with frequency enhancement (MSFF-FE) module, frequency-focused down-sampling (FD), semantic alignment and calibration (SAC) module, illustrated in Figure 3 (b.2).

## 4.4 Hierarchical Progressive Decoder

We propose the hierarchical progressive decoder that utilizes multi-level language and visual features to guide object queries to decode more relevant spatial information. We show more technical detail in Sec. C.2 of the supplementary material.

**Multi-level Language Modulation Module.** This module is designed to enhance the visually contextualized language features by incorporating different levels of visual features. Given language features $\boldsymbol{F}_l^N = \left\{ \boldsymbol{F}_{lt}^N \in \mathbb{R}^{C \times N_l} \right\}_{t=1}^{T}$ and multi-level visual features $\boldsymbol{F}_{vi}^F = \left\{ \boldsymbol{F}_{vit}^F \in \mathbb{R}^{C \times N_{vi}} \right\}_{t=1}^{T}$,

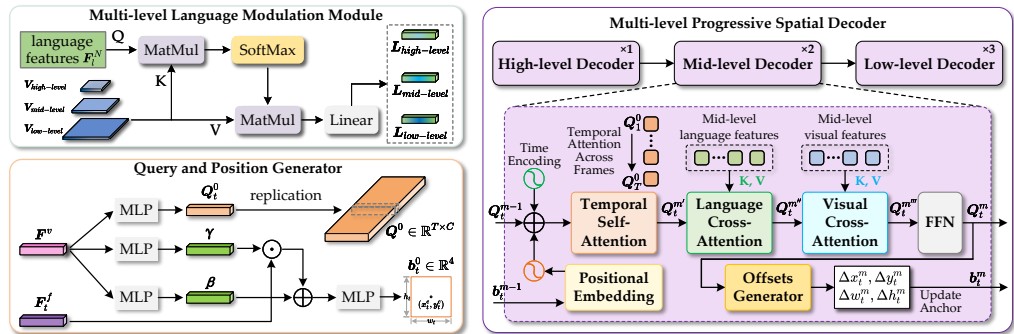

Figure 4: The structure of the multi-level language modulation module, the query and position generator, and the multi-level progressive spatial decoder.

we employ cross-attention to achieve cross-modality fusion:

$$\text{Attn}_t^i = \left( \frac{proj_q(\boldsymbol{F}_{lt}^N)proj_k(\boldsymbol{F}_{vit}^F)^T}{\sqrt{d}} \right), \tag{1}$$

$$\boldsymbol{F}_{lt}^i = \boldsymbol{W}_l \left[ \text{SoftMax}(\text{Attn}_t^i) \cdot proj_v(\boldsymbol{F}_{vit}^F) \right] + \boldsymbol{b}_l, \tag{2}$$

where $proj_{q,k,v}$ are the query, key, and value projections, $\boldsymbol{W}_l$ and $\boldsymbol{b}_l$ are the learnable parameters, and $\boldsymbol{F}_{lt}^i \in \mathbb{R}^{C \times N_l}$ denotes the updated $i$-th level language features in the $t$-th frame. The resulting multi-level language features at different levels are:

$$\boldsymbol{L}_{low-level}, \boldsymbol{L}_{mid-level}, \boldsymbol{L}_{high-level} = \left\{ \boldsymbol{F}_{lt}^3 \right\}_{t=1}^T, \left\{ \boldsymbol{F}_{lt}^4 \right\}_{t=1}^T, \left\{ \boldsymbol{F}_{lt}^5 \right\}_{t=1}^T. \tag{3}$$

**Query and Position Generator.** Existing advanced encoder-decoder based grounding methods either use learnable embeddings [17, 1, 19] or use language features [2, 20] as decoder queries. Despite achieving advanced results in natural scenes, these methods in complex aerial scenes exacerbate the problem of inconsistent localization results across frames. To alleviate this problem, we leverage video tokens $\boldsymbol{F}^v$ and frame tokens $\boldsymbol{F}^f$ in the intra-scale multi-modality interaction branch to generate queries and predict the initial bounding box tube as reference anchors. The structure is shown in Figure 4. Firstly, we project the video tokens $\boldsymbol{F}^v$ as the initial queries $\boldsymbol{Q}_t^0 \in \mathbb{R}^C$. The $\boldsymbol{Q}_t^0$ is temporally replicated $T$ times for each frame resulting in object queries $\boldsymbol{Q}^0 = \left\{ \boldsymbol{Q}_t^0 \right\}_{t=1}^T$. Unlike the previous approach, our queries are mapped from video tokens. All frames are associated and contain the same vision-language contextualized semantics. Such a mechanism helps to make spatial positions of the decoding more temporally consistent.

Then, we encode video tokens to modulate the previous frame tokens $\boldsymbol{F}^f$ by scaling and shifting to generate the initial reference anchor per frame $\boldsymbol{B}^0 = \left\{ \boldsymbol{b}_t^0 \right\}_{t=1}^T$. Specifically, video tokens $\boldsymbol{F}^v$ are projected as a scaling factor $\gamma \in \mathbb{R}^C$ and a shifting factor $\beta \in \mathbb{R}^C$ respectively:

$$\gamma = \tanh\left( \boldsymbol{W}_\gamma \boldsymbol{F}^v + \boldsymbol{b}_\gamma \right), \beta = \tanh\left( \boldsymbol{W}_\beta \boldsymbol{F}^v + \boldsymbol{b}_\beta \right), \tag{4}$$

where $\boldsymbol{W}_\gamma$, $\boldsymbol{b}_\gamma$, $\boldsymbol{W}_\beta$, and $\boldsymbol{b}_\beta$ are the learnable parameters. Afterwards, frame tokens are then refined with the two modulation factors. Finally, generate the initial reference anchor box $\boldsymbol{b}_t^0 \in \mathbb{R}^4$:

$$\boldsymbol{b}_t^0 = \text{Sigmoid}\left[ \boldsymbol{W}_p \left( \boldsymbol{F}_t^f \odot \gamma + \beta \right) + \boldsymbol{b}_p \right], \tag{5}$$

where $\odot$ represents hadamard product, and $\boldsymbol{W}_p \in \mathbb{R}^{C \times 4}$ and $\boldsymbol{b}_p \in \mathbb{R}^4$ are learnable parameters.

**Multi-level Progressive Spatial Decoder.** Existing DETR decoders project object queries to bounding box coordinates via a prediction head. It is difficult to accurately and consistently ground the referred object across frames in aerial video with small objects and complex motion. To improve the grounding consistency, we propose the multi-level progressive spatial decoder (MPSD). Previous object detectors [56, 57] amplify the importance of low-level features to enhance small object detection. Therefore, MPSD decodes from high level to low level, and gradually increases the number

of decoding layers for lower-level features. As shown in Figure 4, our MPSD has $M$ layers and is cascaded by 1 High-level, 2 Mid-level, and 3 Low-level Decoders. First, reference anchor boxes are attached to object queries through positional embedding, guiding queries to capture the spatial information of the referred object:

$$P_t^m = \text{MLP}(\text{SinEmbed}(b_t^{m-1})), \tag{6}$$

where SinEmbed means the sinusoidal position encoding to $b_t^{m-1} = (x_t^{m-1}, y_t^{m-1}, w_t^{m-1}, h_t^{m-1})$. In addition to positional embedding $P_t^m$, sinusoidal temporal positional encoding is added to the positional part of object queries of the $m$-th layer. Then, object queries $Q^m$ perform long temporal dependency modeling along the temporal dimension and progressively decode the object position from high-level to low-level under the guidance of multi-level language and visual features:

$$Q_t^{m'} = \text{LN}\left(Q_t^{m-1} + \text{MHSA}_{temporal}^m(Q_t^{m-1})\right), \tag{7}$$

$$Q_t^{m''} = \text{LN}\left(Q_t^{m'} + \text{MHCA}_{language}^m(Q_t^{m'}, L, L)\right), \tag{8}$$

$$Q_t^{m'''} = \text{LN}\left(Q_t^{m''} + \text{MHCA}_{visual}^m(Q_t^{m''}, V, V)\right), \tag{9}$$

$$Q_t^m = \text{LN}\left(Q_t^{m'''} + \text{FFN}^m(Q_t^{m'''})\right), \tag{10}$$

where $\text{LN}(\cdot)$ denotes layer normalization. The multi-level language and visual feature inputs at different layers are:

$$(L, V) = \begin{cases} (L_{high-level}, V_{high-level}) & \text{if } m = 1 \\ (L_{mid-level}, V_{mid-level}) & \text{if } m = 2 \text{ or } 3 \\ (L_{low-level}, V_{low-level}) & \text{if } m = 4, 5 \text{ or } 6. \end{cases} \tag{11}$$

Finally, we fed $Q_t^m$ into the offsets generator consisting of 3 fully connected layers with the ReLU activation function. The offsets generator directly regresses 4-dim bounding box offset coordinates $(\Delta x_t^m, \Delta y_t^m, \Delta w_t^m, \Delta h_t^m)$. The final reference anchor box $b_t^m$ is updated by:

$$b_t^m = (x_t + \Delta x_t^m, y_t + \Delta y_t^m, w_t + \Delta w_t^m, h_t + \Delta h_t^m). \tag{12}$$

Our improved spatial grounding loss function is in Sec. C.3 of the supplementary material.

## 5 Experiments

In this section, we conduct extensive experiments to verify our SAVG-DETR. We first introduce implementation details and the evaluation protocol for the SAVG task in Sec. D.1 of the supplementary material and Sec. 5.1. After this, we compare with the state-of-the-art methods in Sec. 5.2. We perform extensive ablation studies to investigate the effect of each component of SAVG-DETR in Sec. 5.3. Finally, we visualize some examples for an intuitive understanding of the approach in Sec. 5.4.

### 5.1 Evaluation Metrics

We follow the literature [17] and define the video intersection over union as $vIoU = \frac{1}{N_f} \sum_{t=1}^{N_f} IoU(\hat{b}_t, b_t)$, where $N_f$ represents the total number of frames of the video. $\hat{b}_t$ and $b_t$ are the predicted and ground-truth boxes at time $t$, respectively. To evaluate spatial aerial video grounding, we employ **m_vIoU** and **vIoU@R** as evaluation criteria. m_vIoU is the average $vIoU$ of video samples. The prediction for a video is considered "accurate" if $vIoU$ exceeds a threshold. vIoU@R is the proportion of video samples for which $vIoU > R$. The above metrics evaluate the model's performance based on the global spatial localization accuracy of the video. To assess the model's localization stability for each frame, we introduce a novel metric, $fAcc$. The prediction for a frame in the video is considered "accurate" if the frame IoU exceeds 0.5. $fAcc$ represents the proportion of frames in a video that are predicted correctly. **m_fAcc** denotes the average $fAcc$ across all videos samples. **fAcc@R** is the proportion of video samples where $fAcc > R$. The threshold $R$ is usually set to 0.3 and 0.5 during testing.

Table 2: Performance comparisons of the state-of-the-art methods on the UAV-SVG test set.

| Methods | Visual Encoder | Language Encoder | m_vIoU | vIoU@0.3 | vIoU@0.5 | m_fAcc | fAcc@0.3 | fAcc@0.5 |
|---|---|---|---|---|---|---|---|---|
| Co-grounding [35]CVPR'21 | Darknet53 | Bi-LSTM | 10.24 | 21.66 | 6.11 | 11.17 | 16.29 | 8.40 |
| DCNet [36]ACMMM'22 | Darknet53 | BERT | 11.65 | 23.58 | 8.79 | 13.10 | 17.64 | 9.21 |
| TubeDETR [17]CVPR'22 | ResNet101 | RoBERTa | 22.60 | 32.91 | 20.49 | 23.84 | 29.69 | 22.00 |
| STCAT[18]NeurIPS'22 | ResNet101 | RoBERTa | 24.14 | 35.51 | 22.48 | 27.17 | 33.39 | 25.36 |
| SGFDN [45]ACMMM'23 | ResNet101 | RoBERTa | 20.13 | 28.16 | 15.47 | 19.13 | 22.71 | 17.39 |
| CG-STVG [19]CVPR'24 | ResNet101 | RoBERTa | 21.23 | 28.82 | 19.04 | 22.32 | 26.24 | 20.41 |
| VideoGrounding-DINO [20]CVPR'24 | Swin-Trans. | BERT | 23.83 | 33.84 | 19.92 | 25.80 | 31.72 | 23.00 |
| **SAVG-DETR** (Ours) | ResNet101 | RoBERTa | **27.15** | **38.18** | **22.85** | **28.82** | **35.85** | **26.55** |

Table 3: Ablation study of key components of our SAVG-DETR framework.

| Encoder | | Decoder | | | m_vIoU | m_fAcc |
|---|---|---|---|---|---|---|
| IMIB | CVFB | MLMM | QPG | MPSD | (%) | (%) |
| ✓ | | | | | 22.38 | 25.46 |
| ✓ | | | | ✓ | 19.37 | 21.84 |
| ✓ | ✓ | | | ✓ | 25.44 | 26.54 |
| ✓ | ✓ | ✓ | | ✓ | 26.88 | 27.92 |
| ✓ | ✓ | ✓ | ✓ | ✓ | **27.15** | **28.82** |

Table 4: Comparisons of the variants and baselines.

| Variants or Methods | m_vIoU (%) | m_fAcc (%) | FLOPs (G) | Param (M) | Mem (G) |
|---|---|---|---|---|---|
| A | 17.64 | 18.93 | 65.07 | 184.22 | 9.8 |
| B | 22.11 | 24.30 | 120.51 | 199.77 | 14.4 |
| C | / | / | / | 199.77 | >48 |
| D | 26.64 | 27.49 | 245.37 | 215.92 | 31.5 |
| E (**Ours**) | 27.15 | 28.82 | 203.08 | 209.23 | 28.7 |
| TubeDETR [17] | 22.60 | 23.84 | 144.04 | 185.17 | 11.5 |
| STCAT [18] | 24.14 | 27.17 | 175.46 | 207.14 | 15.9 |
| SGFDN [45] | 20.13 | 19.13 | 53.65 | 178.91 | 4.3 |
| CG-STVG [19] | 21.23 | 22.32 | 231.31 | 192.83 | 27.3 |

## 5.2 Comparison with the State-of-the-art Methods

To fully verify the superiority of our proposed SAVG-DETR, we compare it with all SOTA methods on Table 2. Specifically, we provide two sets of comparison methods: 1) SOTA video REC methods: Co-grounding [35] and DCNet [36]. 2) SOTA spatio-temporal video grounding methods: TubeDETR [17], STCAT [18], SGFDN [45], CG-STVG [19], and VideoGrounding-DINO [20]. To date, CG-STVG and VideoGrounding-DINO have achieved the best performance in natural scenes. Our SAVG-DETR outperforms the state-of-the-arts consistently in all evaluation metrics. We provide more detailed baselines, result analyses, advantages and disadvantages of different methods in Sec. D.2 of the supplementary material.

## 5.3 Ablation Studies

**Ablation study on key components of SAVG-DETR.** In Table 3, we conduct a thorough ablation study on the proposed components. The first row performs SAVG with only intra-scale multi-modality interaction branch (IMIB), vanilla decoder, and prediction head. On this basis, we further introduce multi-scale visual features into the decoder. It can be found that the accuracy drops significantly by about 3 points, in the second row of Table 3. IMIB only processes high-level single-scale interaction, while other lower-level visual features without multi-modal interactions cannot be accurately decoded. To solve this problem, in the third row, we further add the cross-scale visual-only fusion branch (CVFB) to achieve multi-scale fusion. Lower-level visual features gather contextual information from the high-level visual features of multi-modal interactions. We find the accuracy is improved by about 6 points. The fourth row modulates language features through multi-scale visual features and guides object queries to capture spatial information more accurately in the decoder. We find the accuracy is again boosted by about 1 point. The last row shows that after applying the query and position generator (QPG), our full-fledged model achieves the best performance.

**Design of the multi-modality multi-scale spatio-temporal encoder and computation analysis.** In Table 4, we evaluate the performance and complexity of the variants designed in Sec. C.1 of the supplementary material and other baselines. Compared to variant A, variant B has an approximately 5% increase in performance and an 85% increase in FLOPs. This proves that intra-scale multi-modal interaction is very important, but the Transformer encoder has a high computational overhead. The computational cost of the self-attention is shown as a quadratic increase in the sequence length of the input. Variant C maintains the same parameter size as B, but it inputs multi-scale multi-modal features of long sequences, resulting in a significant increase in FLOPs and an unaffordable memory demand (out of memory). Variant D reduces FLOPs compared with C and has a performance increase of about 3.5% over B, indicating that our decoupling strategy not only reduces computational complexity but also increases performance. Our SAVG-DETR offers 0.9% performance improvement and 17% FLOPs reduction over D. Compared with other methods, SAVG-DETR has the largest size and memory usage due to the processing of multi-scale features. However, we efficiently decouple multi-modality and multi-scale spatio-temporal modeling, which significantly improves performance.

Table 5: More ablation studies. Detailed analysis is shown in Sec. D.3 of the supplementary material.

(a) Intra-scale multi-modality interaction branch

| Video tokens | SIL | TIL | m_vIoU | m_fAcc |
|---|---|---|---|---|
| Average pooled frame tokens | ✓ | | 25.21 | 25.58 |
| Max pooled frame tokens | ✓ | | 25.92 | 27.14 |
| Learnable tokens | ✓ | ✓ | **27.15** | **28.82** |

(b) Object query initialization strategies

| Query generation | m_vIoU | m_fAcc |
|---|---|---|
| Zero vector embedding | 26.94 | 28.51 |
| Average pooled language tokens | 24.91 | 25.11 |
| Max pooled language tokens | 25.37 | 25.96 |
| Query and position generator | **27.15** | **28.82** |

(c) Time encoding and temporal self-attention on the MPSD

| Time Encoding | Temporal Self-Attention | m_vIoU (%) | m_fAcc (%) |
|---|---|---|---|
| | | 25.56 | 25.30 |
| ✓ | | 25.95 | 26.66 |
| | ✓ | 26.21 | 27.79 |
| ✓ | ✓ | **27.15** | **28.82** |

(d) Multi-level vision-language features on the MPSD

| Language features | Visual features | m_vIoU (%) | m_fAcc (%) |
|---|---|---|---|
| | | 23.12 | 25.79 |
| ✓ | | 24.20 | 25.90 |
| | ✓ | 25.96 | 26.79 |
| ✓ | ✓ | **27.15** | **28.82** |

(e) Positional embedding and offsets generator on the MPSD

| Positional Embedding | Offsets Generator | m_vIoU (%) | m_fAcc (%) |
|---|---|---|---|
| learnable | | 24.84 | 24.42 |
| learnable | ✓ | 22.99 | 24.49 |
| reference anchors | | 23.57 | 24.18 |
| reference anchors | ✓ | **27.15** | **28.82** |

(f) Number of decoder layers on the MPSD.

| Layer Index | M = 4 | | M = 5 | | M = 6 | | M = 7 | |
|---|---|---|---|---|---|---|---|---|
| | m_vIoU | m_fAcc | m_vIoU | m_fAcc | m_vIoU | m_fAcc | m_vIoU | m_fAcc |
| $m = 7$ | - | - | - | - | - | - | 26.88 | 28.21 |
| $m = 6$ | - | - | - | - | **27.15** | **28.82** | 26.82 | 28.16 |
| $m = 5$ | - | - | 25.94 | 27.35 | 26.93 | 28.45 | 25.65 | 27.34 |
| $m = 4$ | 21.98 | 23.01 | 25.12 | 26.04 | 25.82 | 27.68 | 25.16 | 26.65 |
| $m = 3$ | 21.36 | 22.23 | 24.53 | 25.33 | 25.94 | 26.38 | 24.58 | 25.30 |
| $m = 2$ | 20.89 | 21.77 | 23.26 | 24.93 | 24.48 | 25.55 | 23.39 | 23.64 |
| $m = 1$ | 19.45 | 20.27 | 22.48 | 23.94 | 22.87 | 24.09 | 21.32 | 22.81 |

(g) Different level decoder combinations.

| High-level Decoder | Mid-level Decoder | Low-level Decoder | m_vIoU (%) | m_fAcc (%) |
|---|---|---|---|---|
| 6 | 0 | 0 | 23.12 | 25.79 |
| 3 | 2 | 1 | 24.38 | 26.23 |
| 2 | 2 | 2 | 26.24 | 27.47 |
| 1 | 2 | 3 | **27.15** | **28.82** |
| 1 | 1 | 4 | 25.36 | 26.71 |
| 0 | 0 | 6 | 22.89 | 23.67 |

**Expression 1:** The only white speedboat towards the upper left of the sea, sailing with a rubber dinghy.

**Expression 2:** The only white coach that is driving near the building in the lower right corner, towards the left.

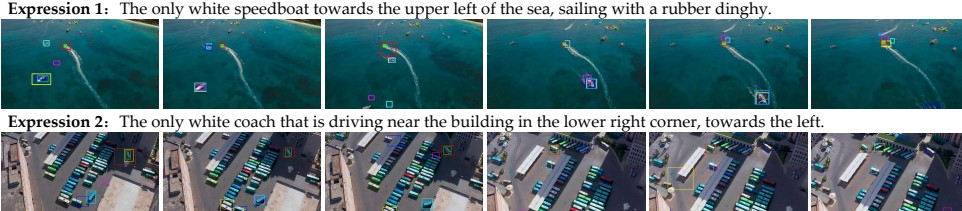

Ground Truth   SAVG-DETR   TubeDETR   STCAT   SGFDN   CG-STVG   VideoGrounding-DINO

Figure 5: Qualitative results of different methods on the UAV-SVG benchmark.

SAVG-DETR achieves a better trade-off between performance and complexity. We provide more detailed ablation studies in Sec. D.3 of the supplementary material.

## 5.4 Visualization Analysis

In Figure 5, we present some qualitative examples, comparing with other methods. The challenge posed by small objects in aerial scenes is intuitively apparent. Additionally, in **Expressions 1** and **2**, multiple small, similar moving objects of the same class appear in frames, presenting a significant challenge. In **Expressions 1** and **2**, it is necessary to understand both the object semantic and the complex spatial motion, and to reason about the regional location of the speedboat and coach. In **Expression 2**, although STCAT (yellow) can initially detect the coach in the first frame, it gradually fail to locate it. Our proposed SAVG-DETR (red) performs well and achieves reasonable localization results. SGFDN (cyan) locates the blue coach as it pulls into the building in the lower right corner. As the drone moves, the blue coach disappears from view and SGFDN locates the parked white coach. We provide more qualitative results and failure analysis in Sec. D.4 of the supplementary material,

## 6 Conclusion

In this paper, we introduce a novel SAVG task and contribute a challenging large-scale benchmark UAV-SVG. To improve the grounding performance of aerial small objects and consistency across frames, we propose SAVG-DETR framework. The core design is a multi-modality multi-scale spatio-temporal encoder for cross-modal cross-scale feature fusion and alignment, and a hierarchical progressive decoder for efficient spatial tube prediction. From coarse to fine, SAVG-DETR gradually bridges the modality gap and iteratively refines reference anchors of the referred object. Extensive experiments validate the effectiveness and superiority of the proposed method.

# Acknowledgments

This work is supported in part by grants from the National Key Research and Development (R&D) Program of China (No.2024YFC3015504) and the Basic Research Project for Young Students of the National Natural Science Foundation of China (No.624B2113).

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

## Technical Appendices and Supplementary Material

In the supplementary material, we will introduce the following content to complement the details of our study.

- Sec. A: More Introduction
- Sec. B: More Benchmark Details
- Sec. C: More Details of the Methodology
- Sec. D: More Experiments
- Sec. E: Licenses
- Sec. F: Limitations and Future Work
- Sec. G: Societal Impact

# A    More Introduction

We provide some data samples of existing benchmarks in natural scenes, and our UAV-SVG benchmark samples are shown in Figure 6. The SVG task in natural scenes has predominantly focused on the ego-centric fixed front perspective and a simple scene, which only provides a very limited view and environment. The referred object covers a significant portion of the frame image. Moreover, the main objects referred to are mainly human beings.

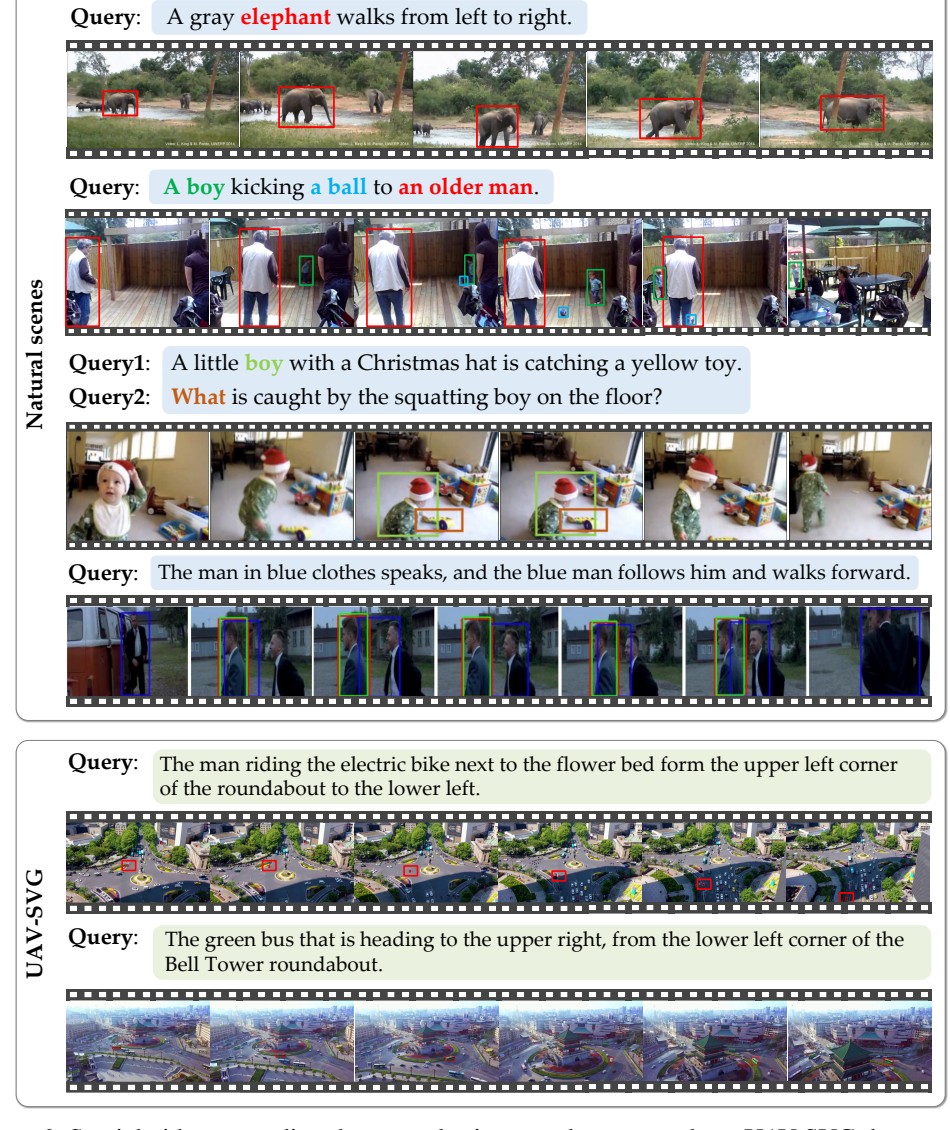

Figure 6: Spatial video grounding data samples in natural scenes and our UAV-SVG data samples.

# B More Benchmark Details

## B.1 Dataset Annotation

The construction pipeline of the newly proposed dataset is shown in Figure 2. **Step 1: Manual Data Cleaning and Spatio-Temporal Tube Correction.** Due to the hard defects of WebUAV-3M, such as signal instability, black screen, sudden change of inter-frame resolution, and target out of view, we carried out data cleaning and spatio-temporal tube clipping. The temporal tubes and bounding boxes of the video sequences are verified manually to ensure accuracy. Referring to VID-sentence [6], we delete videos with less than 9 frames. **Step 2: Referring Expression Generation.** We design the object-specific prompt with human priors to generate referring expressions by the Gemini 1.5 Pro model. Specifically, we incorporate WebUAV-3M's manually annotated object descriptions into the prompt. The Gemini 1.5 augments more referring expressions with reference to the keyframe images and human prior information. Due to Gemini's limitation in continuous input processing, we integrate keyframes into a single composite image for input. Gemini 1.5 Pro is capable of handling contexts of up to 1 million tokens, which is currently the longest context window of any large model. In addition, many studies [58, 59] and reviews have shown that the Gemini 1.5 Pro performs better than GPT-4o. **Step 3: Manual Checking and Error Fixing.** Our team strives to maintain non-ambiguous and high-quality annotation through manual quality control. Each expression is manually checked to determine whether the described attributes are correct and whether the referred object can be uniquely distinguished. Correct any errors that exist in the raw expressions. If an instance is difficult to describe uniquely and precisely or is hard to distinguish from other objects, discard this sample. Our manual verification of the dataset takes about 3 months.

## B.2 Analyses of UAV-SVG

This section analyzes the salient differences between UAV-SVG and existing video grounding benchmarks. The three most widely used video-based spatial visual grounding datasets for natural scenes include VID-sentence [6], VidSTG [7], and HC-STVG [8]. Our analysis encompasses the following viewpoints.

**Video resolution and scales of bounding boxes.** In Figure 7 (a), the distribution pattern of the video resolution in different benchmarks is illustrated by the different color circle distributions and the area of circles. It is evident that UAV-SVG's video resolution (red) is more widely distributed and contains more high-resolution videos than other datasets. The natural scene video is mainly concentrated in the area of 1,200 × 1,000. UAV-SVG contains more video than this resolution, even up to 2,000 pixels wide or high. In Figures 7 (b), (c), and (d), the relative area and absolute area of the bounding box are shown respectively. It is clear that UAV-SVG exhibits greater scale differences compared to other datasets. The relative area and absolute area of the bounding box in UAV-SVG are small, and the absolute area is mainly concentrated within 200 pixels. This data reveals the challenges of small object grounding for the UAV-SVG benchmark.

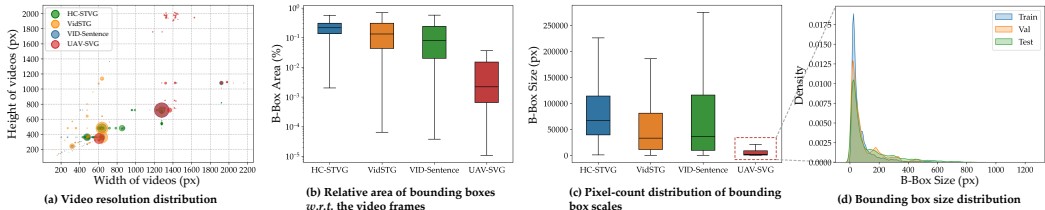

Figure 7: A comparative analysis of our UAV-SVG against the HC-STVG, VidSTG, and VID-Sentence benchmarks. (a) represents the distribution of video resolution (height and width), and the area of each circle is proportional to the number corresponding resolution. (b) and (c) illustrate the distribution patterns of the relative area of each frame bounding box with respect to the frame size and the pixel count (in terms of the product of height and width). (d) shows the density distribution map of the bounding box size of our UAV-SVG dataset in detail.

**Lengths of expressions.** As shown in Table 1, the descriptions of UAV-SVG contain a vocabulary of 3,242 words. The minimum and maximum length of the sentences are 4 and 46, respectively. The descriptions have an average of 16.39 words. We demonstrate the word count distribution of the expression lengths in Figures 8 (a) and (b). We note that the expression lengths of VID-sentence and VidSTG are shorter and are mainly distributed to the left of 15. HC-STVG and our UAV-SVG are much more widely distributed. Figure 8 (c) shows the word clouds for all descriptions of UAV-SVG. We can see that UAV-SVG covers a wide range of descriptions, including objects, attributes, relationships, motions, etc. Longer expressions can accommodate more details about the object's attributes, appearance, location, relationships, and changes, which increases the semantic space that the model needs to consider and brings challenges to spatial video grounding.

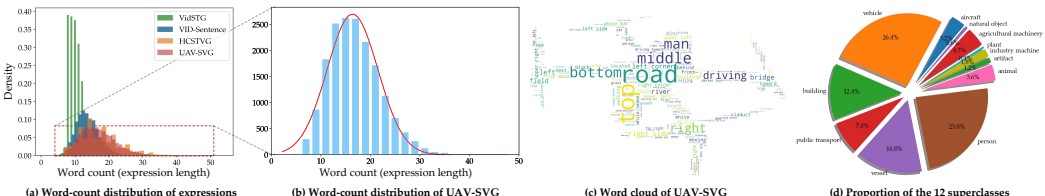

**(a) Word-count distribution of expressions**     **(b) Word-count distribution of UAV-SVG**     **(c) Word cloud of UAV-SVG**     **(d) Proportion of the 12 superclasses**

Figure 8: (a) and (b) show the sentence lengths' distribution. (c) presents the word cloud of UAV-SVG vocabulary with each word proportional to its frequency. (d) shows the proportion of each object superclass in UAV-SVG.

**Quantities of object classes.** As shown in Figure 8 (d), all videos of UAV-SVG are divided into 12 superclasses, including person, animal, vehicle, building, vessel, public transport, aircraft, agricultural machinery, industry machine, plant, artifact, and natural object. More specifically, UAV-SVG includes 216 object classes and 73 motion classes in total. The histograms of the object classes and the motion classes are shown in Figures 9 and 10, respectively. We can observe that the entire aerial videos and the number of videos in each set of superclasses present a long-tail distribution. For example, the vehicle and person superclasses contain 941 and 849 videos, respectively, while the natural object and plant superclasses only have 21 and 16 videos. For example, the sedan and SUV objects in the vehicle superclass contain 210 and 164 videos, respectively, while the police van has only 1 video. These reflect the true distribution of objects in the aerial video source. As shown in Table 1, our UAV-SVG clearly has more diverse and richer object categories than other natural scene datasets. These long-tail distributions and rich object classes pose significant challenges in building accurate and robust grounding models for aerial video.

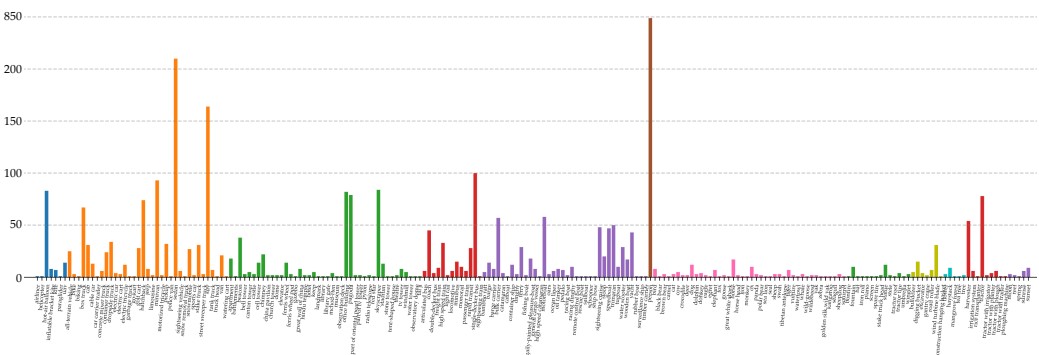

Figure 9: The number of videos per group of object classes. The colors of the bar chart represent 12 superclasses, from left to right: aircraft, vehicle, building, public transport, vessel, people, animal, artifact, agricultural machinery, and natural object. The detailed classification of the people superclass is shown in Figure 10. Best viewed by zooming in.

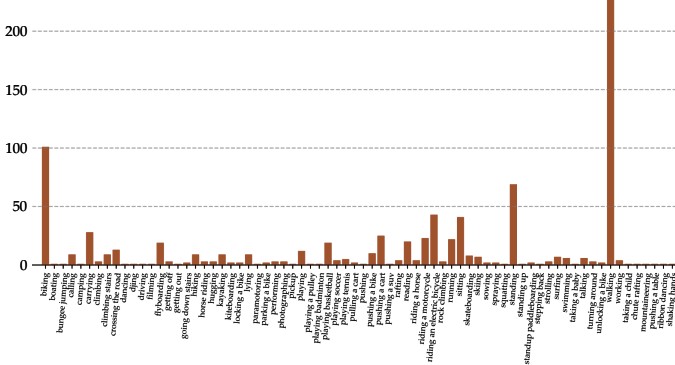

Figure 10: The number of videos per group of motion classes. Best viewed by zooming in.

**Target position distribution.** The distribution of the normalized target center position in different datasets is shown in Figure 11. HC-STVG's videos come from movies and are carefully selected with human-centric clips. The object centers of HC-STVG are clearly distributed around y=0.5 and below. Because movies are mainly eye-level line shots, that is, the camera is roughly parallel to the line of sight of the subject, and the character is placed in the middle of the picture. The videos of VID-sentence and VidSTG are both hand-held or ego-centric fixed front perspectives, which provide only a very limited view. VidSTG is radial in three directions from the center point to the left, right, and down. The position distribution of VID-sentence radiates from the center to the periphery, and UAV-SVG is similar. In addition, UAV-SVG also contains a number of scattered objects distributed in the edge parts. In Figure 11, the targets of UAV-SVG in the training and test sets have similar position distributions, concentrated (*i.e.*, highlighted) in the central region of the images.

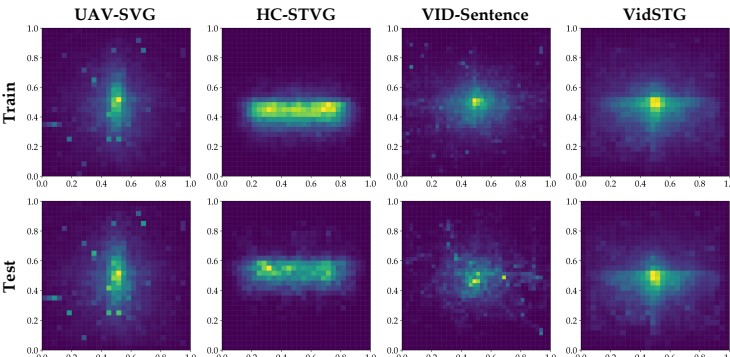

Figure 11: Target position distributions. Best viewed by zooming in.

**Other statistics** UAV-SVG contains over 2.01 million frames across 3,564 videos and offers 216 highly diverse object categories. The total duration and average duration of videos are 18.7h and 18.86s, as shown in Table 1. There are 17,820 video-sentence-tube triples in our newly constructed UAV-SVG dataset, including 14,060 training triples, 845 validation triples, and 2,915 testing triples, shown in Table 6. We believe that the unique characteristics of UAV-SVG can open the door for the spatial aerial video grounding paradigm with practically useful and broader real-life applications.

Table 6: Dataset Statistics of the UAV-SVG dataset.

|       | #Query | #Video | #Super. | #Object. | #Motion. |
|-------|--------|--------|---------|----------|----------|
| Train | 14,060 | 2,812  | 12      | 202      | 67       |
| Val   | 845    | 169    | 10      | 50       | 14       |
| Test  | 2,915  | 583    | 12      | 113      | 36       |
| All   | 17,820 | 3,564  | 12      | 216      | 73       |

## B.3 Dataset Characteristics

Spatio-temporal video grounding in natural scenes is quite different from that in aerial scenes. As shown in Table 1, the scale of referring expressions in VID-sentence and HC-STVG is relatively small. Moreover, the objects of VID-sentence are limited to two superclasses, animal and vehicle. HC-STVG is a human-centric video dataset with only humans as its objects. VidSTG had the largest data size, but the proportion of objects in human and animal superclasses reaches 92.12%. In summary, the scenarios of the above datasets are simple and the number of object categories is limited. In contrast, UAV aerial videos have a broader field of view and encompass a wider and more complex range of object categories, posing a higher challenge to the spatio-temporal video grounding task. The top left corner of Figure 2 illustrates some challenging samples. Specifically, compared to existing datasets, our proposed UAV-SVG dataset has the following characteristics and challenges:

1. *Fast Motion (FM):* The motion of the object is too fast, resulting in large bounding box differences between each frame, such as the wind turbine blade.

2. *Illumination Variations (IV):* The illumination of the target region changes due to the sun irradiation and the movement of the camera. There is also a low-illumination problem at dusk and night.

3. *Partial Occlusion (PO):* The object is partially occluded in the video sequence, such as the truck entering the interior of the viaduct.

4. *Camera Motion (CM):* The UAV carrying the camera may suddenly move and cause the picture to change rapidly.

5. *Scale Variations (SV):* The object bounding box ratio varies greatly between different frames, such as a ship in the frame.

6. *Viewpoint Changes (VC):* Due to the co-motion of the camera and the object, the viewpoint changes, which seriously affects the appearance of the object. For example, a cable car that moves on a cable.

7. *ROTation (ROT):* Objects continuously rotate in the video sequence, such as a car driving on a curved highway.

8. *Aspect Ratio Variations (ARV):* The aspect ratio of the object bounding box varies greatly, such as a high-speed train running at high speed.

9. *Low Resolution (LR):* Due to the large sky view field of UAV, most of the objects in aerial video are small targets with low resolution, especially the human and car.

10. *DEFormation (DEF):* Objects are deformable during tracking, such as the player in intense motion on the basketball court.

11. *Background Clutter (BC):* In the scene of aerial photography, the background and the object often have similar appearance, which is easy to be confused and difficult to distinguish.

12. *Motion Blur (MB):* The object region is blurred by the fast motion of the target or the camera, such as a fast flying white bird.

## C   More Details of the Methodology

As shown in Figure 3, our proposed SAVG-DETR consists of an aerial video-text feature extractor, a multi-modality multi-scale spatio-temporal encoder, and a hierarchical progressive decoder. In this section, we present a more concrete exposition for our multi-modality multi-scale spatio-temporal encoder (Sec. C.1), hierarchical progressive decoder (Sec. C.2), and improved spatial grounding loss function (Sec. C.3).

### C.1   Multi-Modality Multi-Scale Spatio-Temporal Encoder

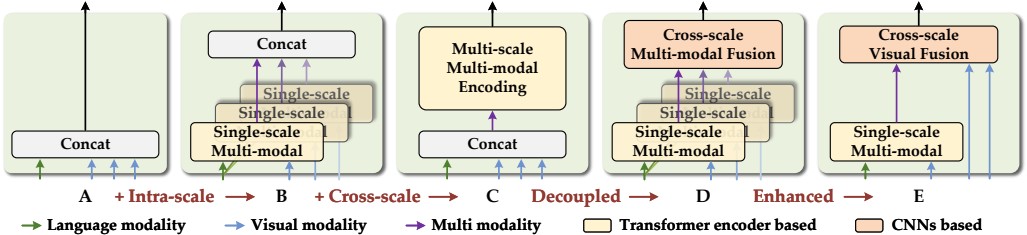

Figure 12: The different types of multi-modal multi-scale encoder variants and the evolution process. The number of blue arrows indicates the number of visual scales.

The introduction of multi-scale features in UAV object detection can accelerate convergence and improve performance [53, 54]. Inspired by this, we introduce multi-scale visual features into the traditional multi-modality spatio-temporal encoder. However, computing the self-attention between multi-scale visual features and language features for each frame is computationally expensive and unaffordable. To overcome this problem, we rethink the structure of the spatio-temporal encoder. In order to intuitively understand the design idea, we show the different types of encoder variants and the evolution process in Figure 12. The details are as follows:

- The initial variant A directly connects multi-modal and multi-scale features without any encoding or fusion, which will not be effectively used for decoding.

- Variant B inserts a single-scale multi-modal encoder to first conduct intra-scale multimodal encoding, followed by feature concatenation for output.

- Variant C directly feeds concatenated multi-modal multi-scale features into a transformer encoder, performing cross-scale cross-modal interaction.

- Variant D decouples multi-scale multi-modal interaction into two cascaded processes of intra-scale multi-modal encoding and CNN-based cross-scale multi-modal fusion.

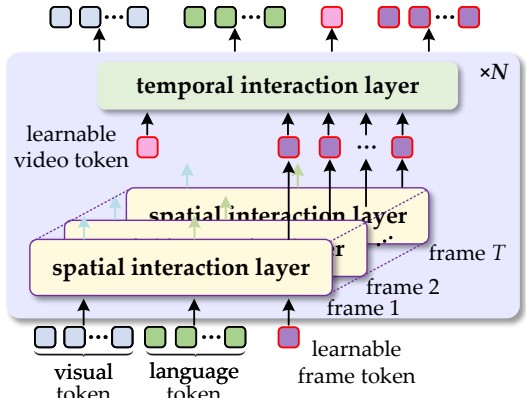

Figure 13: Intra-scale multi-modality interaction branch is a stack of $N$ layers. Each layer consists of spatial interaction layer and temporal interaction layer. The spatial interaction layer fuses information across spatial dimensions and vision-language modalities locally in each frame. The temporal interaction layer fuses information across temporal dimensions in the global video-language context.

- Variant E proposes a more refined and enhanced framework that cascades intra-scale multi-modal interaction and cross-scale visual fusion based on D. Our framework only performs intra-scale multi-modal interaction on $S_5$ and visual-only multi-scale fusion, which further reduces the computational cost of variant D. This is because high-level features contain richer semantic concepts about the object and can capture the referred object entity in conjunction with the language modality. However, multi-modal interaction of low-level features lacking semantic information may introduce confusion or redundancy with high-level multi-modal interactions.

Based on the above analysis, we propose the Intra-Scale Multi-Modality Interaction Branch and the Cross-Scale Visual-only Fusion Branch.

### C.1.1 Intra-scale Multi-modality Interaction Branch

This branch aims to model multi-modality interactions between the language features $\boldsymbol{F}_l$ and high-level aerial video features $\boldsymbol{F}_{v5} = \left\{ \boldsymbol{F}_{v5t} \in \mathbb{R}^{C \times N_{v5}} \right\}_{t=1}^{T}$. Specifically, the branch consists of a $N$ layer encoder based on self-attention. Each layer starts with a spatial interaction layer followed by a temporal interaction layer, as shown in Figure 13. The spatial interaction layer can conduct local spatial attention on each frame. We introduce a learnable embedding $\boldsymbol{F}_t^f \in \mathbb{R}^{C \times 1}$ (namely frame token) in $t$-th frame to capture the spatial context of the referred object through intra-modality and inter-modality interactions. Then, we formulate the joint input tokens $\boldsymbol{x}_t^s$ of the spatial interaction layer on the $t$-th frame as:

$$\boldsymbol{x}_t^s = [\underbrace{\boldsymbol{F}_{v5_t}^1, \boldsymbol{F}_{v5_t}^2, \dots \boldsymbol{F}_{v5_t}^{N_{v5}}}_{video\ tokens\ \boldsymbol{F}_{v5_t}}, \underbrace{\boldsymbol{F}_l^1, \boldsymbol{F}_l^2, \dots, \boldsymbol{F}_l^{N_l}}_{language\ tokens\ \boldsymbol{F}_l}, \boldsymbol{F}_t^f]. \tag{13}$$

After obtaining the input $\boldsymbol{x}_t^s \in \mathbb{R}^{C \times (N_{v5}+N_l+1)}$ as described above, we apply the self-attention across spatial dimensions to embed $\boldsymbol{x}_t^s$. To retain the positional and modality information, we add learnable position encodings to the input $\boldsymbol{x}_t^s$ of each layer. Thanks to the attention mechanism, frame tokens $\boldsymbol{F}^f = \{\boldsymbol{F}_t^f\}_{t=1}^{T}$ fuse information across spatial dimensions of visual and textual modalities. The spatial interaction layer conducts local spatial modeling for each frame but lacks global temporal modeling which is important for the prediction consistency among aerial video frames. To address this issue, we propose the temporal interaction layer which applies the self-attention across temporal dimensions between learnable frames tokens. Similarly, we introduce a learnable embedding $\boldsymbol{F}^v \in \mathbb{R}^{1 \times C}$ (namely video token) to capture the global aerial video-text context. We formulate the joint input tokens $\boldsymbol{x}^{te}$ of the temporal interaction layer as:

$$\boldsymbol{x}^{tem} = [\underbrace{\boldsymbol{F}_1^f, \boldsymbol{F}_2^f, \dots, \boldsymbol{F}_T^f}_{frame\ tokens\ \boldsymbol{F}^f}, \boldsymbol{F}^v], \tag{14}$$

where $\boldsymbol{x}^{tem} \in \mathbb{R}^{C \times (T+1)}$. To retain the temporal information, we add a sinusoidal temporal position encoding to the positional part of the input $\boldsymbol{x}^{tem}$. After the intra-scale multi-modality interaction branch, we split the aerial video features $\boldsymbol{F}_{v5}^N \in \mathbb{R}^{T \times C \times N_{v5}}$ and language features $\boldsymbol{F}_l^N \in \mathbb{R}^{T \times C \times N_l}$ from the output $\boldsymbol{x}^s \in \mathbb{R}^{T \times C \times (N_{v5}+N_l+1)}$ at layer $N$ and split the output $\boldsymbol{x}^{tem}$ at layer $N$ to frame tokens $\boldsymbol{F}^f \in \mathbb{R}^{T \times C}$ and video

tokens $\boldsymbol{F}^v \in \mathbb{R}^{1 \times C}$. Then we input the high-level video features into the cross-scale visual-only fusion branch, the language features into the hierarchical progressive decoder, and the frame tokens and video tokens into the query and position generator.

### C.1.2  Cross-scale Visual-only Fusion Branch

The structure of this branch is illustrated in Figure 14. The flattened multi-scale features $\{\boldsymbol{F}_{v3}, \boldsymbol{F}_{v4}, \boldsymbol{F}_{v5}^N\}$ are restored to the same shape as the feature maps $\{\boldsymbol{S}_3, \boldsymbol{S}_4, \boldsymbol{S}_5\}$ as input. The fusion block including two $1 \times 1$ convolutions and 3 RepBlocks [60] can fuse two adjacent scale features into a new feature. First, to enhance the detection performance of small and occluded objects, we introduce the MSFF-FE module on the large-scale low-level feature $\boldsymbol{F}_{v3}$. This module employs a cross-stage partial strategy to fuse spatial and frequency domain information from multiple scales, thereby maximally preserving the details of small objects. Second, to prevent the loss of critical spatial details during vanilla downsampling, we introduce a frequency-focused down-sampling strategy that preserves dual-domain information. Finally, to ensure that the semantic information of high-level features can be fully transferred to low-level features, we introduce the SAC module. By aligning and fusing scale $\boldsymbol{S}_5$ with $\boldsymbol{S}_3$, we enhance the semantic representation capability of low-level visual features. The detailed calculation process of the MSFF-FE module, FD strategy, and SAC module can be found in the literature [23]. After the cross-scale visual-only fusion procedure, we flatten the fused multi-scale features and harvest multi-level visual features $\boldsymbol{F}_{vi}^F \in \mathbb{R}^{T \times C \times N_{vi}} (i = 3, 4, 5)$. Then we input the multi-level visual features to our hierarchical progressive decoder for grounding object:

$$\boldsymbol{V}_{low-level}, \boldsymbol{V}_{mid-level}, \boldsymbol{V}_{high-level} = \boldsymbol{F}_{v3}^F, \boldsymbol{F}_{v4}^F, \boldsymbol{F}_{v5}^F. \tag{15}$$

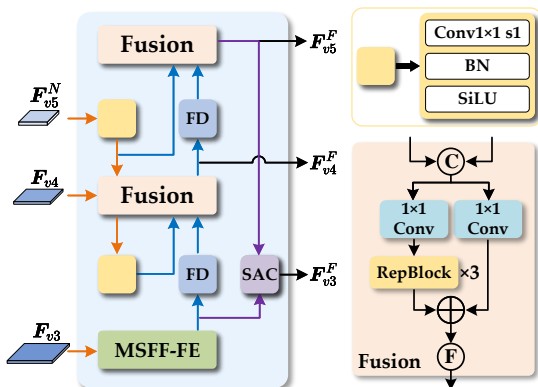

Figure 14: The structure of cross-scale visual-only fusion branch. MSFF-FE denotes the multi-scale feature fusion with frequency enhancement module. FD denotes frequency-focused down-sampling. SAC represents semantic alignment and calibration module.

### C.2  Hierarchical Progressive Decoder

Our proposed hierarchical progressive decoder consists of three submodules: the multi-level language modulation module, the query and position generator, and the multi-level progressive spatial decoder. The modulation module integrates multi-scale visual information into language features. The generator yields context-rich object queries and initial reference anchor boxes. Previous object detectors [56, 57] amplify the importance of low-level features to enhance small object detection capabilities. Therefore, the multi-level progressive spatial decoder decodes from high level to low level, and gradually increases the number of decoding layers for lower-level features. Unlike existing methods, our method does not directly regress the bounding box by cascading a prediction head after the decoder. To improve the prediction consistency, reference anchors as positional embedding constrict the grounding region at each decoder layer. The queries are used to predict offsets and to update the reference anchors at each layer. This framework progressively refines reference anchors of the referred object, eventually grounding the spatial tube.

### C.2.1  Multi-level Language Modulation Module

As shown in Figure 4, the multi-level language modulation module is a cross-attention mechanism with the SoftMax activation function. The resulting multi-level language features at different levels are:

$$\boldsymbol{L}_{low-level}, \boldsymbol{L}_{mid-level}, \boldsymbol{L}_{high-level} = \boldsymbol{F}_l^3, \boldsymbol{F}_l^4, \boldsymbol{F}_l^5, \tag{16}$$

where $\boldsymbol{F}_l^i = \{\boldsymbol{F}_{lt}^i\}_{t=1}^T (i = 3, 4, 5)$. The subsequent spatial decoder can utilize multi-level language features as guidance to make queries decode the more relevant object regions in different levels of visual features.

### C.2.2 Query and Position Generator

The structure is shown in Figure 4. Firstly, we project the video tokens $\boldsymbol{F}^v$ as the initial queries:

$$\boldsymbol{Q}_t^0 = \boldsymbol{W}_q \boldsymbol{F}^v + \boldsymbol{b}_q, \tag{17}$$

where $\boldsymbol{Q}_t^0 \in \mathbb{R}^C$ and $\boldsymbol{W}_q$ and $\boldsymbol{b}_q$ are learnable parameters. The $\boldsymbol{Q}_t^0$ is temporally replicated $T$ times for each frame resulting in object queries $\boldsymbol{Q}^0 = \left\{\boldsymbol{Q}_t^0\right\}_{t=1}^T$. The positional part of the initial queries $\boldsymbol{Q}^0$ is computed using the initial reference anchor boxes $\boldsymbol{B}^0$. For the subsequent decoder layers, these queries and anchor boxes are updated iteratively.

### C.2.3 Multi-level Progressive Spatial Decoder

As shown in Figure 4, each layer consists of six submodules: positional embedding, temporal self-attention, language cross-attention, visual cross-attention, feed-forward network (FFN), and offsets generator.

First, the corresponding positional embedding process of the $m$-th layer is as follows:

$$\boldsymbol{P}_t^m = \text{MLP}(\text{SinEmbed}(\boldsymbol{b}_t^{m-1})). \tag{18}$$

The multi-layer perception (MLP) consists of 2 fully connected layers. In addition to positional embedding $\boldsymbol{P}_t^m$, sinusoidal temporal positional encoding is added to the positional part of the object queries.

Then, the temporal self-attention applies a multi-head self-attention (MHSA) to the object queries along the temporal dimension. The long temporal dependence is modeled by temporal self-attention across frames. Next, based on multi-head cross-attention (MHCA), MPSD utilizes multi-level language and visual features as guidance to progressively decode the object position from high-level to low-level. The lower-level feature contains more spatial details, which is beneficial to spatial grounding. We update the object queries $\boldsymbol{Q}^m$ in each spatial decoder layer.

### C.3 Training Objectives

The ground-truth spatial tube contains the bounding box sequence $\boldsymbol{B} = \{\boldsymbol{b}_t\}_{t=1}^T$. The final predicted box sequence is obtained by $\hat{\boldsymbol{B}} = \{\boldsymbol{b}_t^M\}_{t=1}^T$. The existing literature [17, 19, 20] utilizes the weighted sum of standard $L_1$ loss $\mathcal{L}_{L_1}$ and the Generalized Intersection over Union (GIoU) loss $\mathcal{L}_{GIoU}$ as the spatial grounding loss. However, $\mathcal{L}_{GIoU}$ is less effective for small objects in bounding box regression, especially when the IoU value is low. Inspired by existing work [61], we adopt larger auxiliary bounding boxes to calculate losses, as illustrated in Figure 15. We introduce a scaling-up ratio $r$ to control the width and height of auxiliary ground-truth boxes $\boldsymbol{B}'$ and auxiliary predicted boxes $\hat{\boldsymbol{B}}'$. Formally, the auxiliary IoU is calculated as:

$$\text{Aux-IoU} = \frac{\text{Overlap}}{\text{Union}} = \frac{|\hat{\boldsymbol{B}}' \cap \boldsymbol{B}'|}{|\hat{\boldsymbol{B}}' \cup \boldsymbol{B}'|}. \tag{19}$$

Then, we define the auxiliary GIoU loss as:

$$\mathcal{L}_{AuxGIoU} = \mathcal{L}_{GIoU} + \text{IoU} - \text{Aux-IoU}. \tag{20}$$

Finally, the spatial aerial video grounding loss is defined as:

$$\mathcal{L} = \lambda_{L_1} \mathcal{L}_{L_1}(\hat{\boldsymbol{B}}, \boldsymbol{B}) + \lambda_{AuxGIoU} \mathcal{L}_{AuxGIoU}(\hat{\boldsymbol{B}}, \boldsymbol{B}). \tag{21}$$

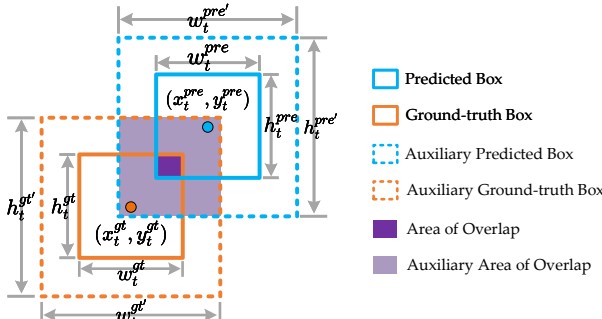

Figure 15: The illustration of larger auxiliary bounding boxes.

# D   More Experiments

## D.1   Implementation Details

To make a fair comparison, we follow the previous work [5, 17, 20] and utilize the ResNet-101 [62] as the visual encoder and RoBERTa [52] as the language encoder. For the transformer-based attention, the number of attention heads is set to 8 and the hidden dimension of feed-forward networks in the attention layer is set to 2,048. We empirically use hyper-parameters $N = 6$, $M = 6$, $\lambda_{L_1} = 5$, and $\lambda_{AuxGIoU} = 4$. We set the initial learning rates to $2 \times 10^{-5}$ for the visual backbone, $5 \times 10^{-5}$ for the language backbone, and $10^{-4}$ for the rest of the network. The learning rate follows a linear schedule with warm-up for the language encoder and the learning rate is dropped by 0.1 after 6 epochs for the rest of the network. We use the AdamW optimizer and weight decay rate $10^{-4}$ for training 20 epochs. Video data augmentation includes spatial random resizing and spatial random cropping preserving box annotations. We sample 5 frames per second for videos and we uniformly sample 100 frames for videos with more than 100 sampled frames. The proposed SAVG-DETR is trained using PyTorch on 2 NVIDIA L20 48G GPUs with 1 video per GPU and the whole optimization takes around 4 days. Due to the limits of GPU memory, the resolution of input on the UAV-SVG dataset is set to 224.

## D.2   Baselines and Quantitative Analysis

**Baselines:**

- Co-grounding [35] and DCNet [36]: They are methods for Video Referring Expression Comprehension, which only use single-frame images and expressions to complete object localization. Co-grounding and DCNet combine adjacent frames to complete object localization.

- TubeDETR [17]: It is the first transformer-based architecture inspired by DETR, specifically including an efficient video-text encoder and a space-time decoder. The encoder models spatial multi-modal interactions over sparsely sampled frames.

- STCAT [18]: To alleviate feature alignment inconsistency and prediction inconsistency, it proposes a novel multi-modal template that explicitly constricts the grounding region and associates predictions between video frames.

- SGFDN [45]: It decomposes 3D spatio-temporal features into 2D motion and 1D object embedding, which can effectively reduce the computational complexity. Based on this strategy, attention is used to capture cross-modal interactions on multiple space-time scales.

- CG-STVG [19]: It learns the discriminant instance context of the object in each decoding stage, serves as a guide to enhance the target awareness in the next decoding stage, and also helps to generate better new instance context to improve the localization.

- VideoGrounding-DINO [20]: It utilizes the pre-trained representations from foundational spatial grounding models to bridge the semantic gap between natural language and diverse visual content, and can effectively respond to open-vocabulary and closed-set scenarios.

**Quantitative Analysis:**

Our SAVG-DETR outperforms the state-of-the-arts consistently in all evaluation metrics. Co-grounding and DCNet are unable to handle T-frame global video features and lack spatio-temporal context modeling. For example, in Figure 3, the bus is heading towards the upper right. Co-grounding and DCNet have lower performance as they fail to provide a global understanding of object space motion.

In particular, we surpass the strong baseline VideoGrounding-DINO on the m_vIoU and m_fAcc metrics with absolute gains of up to 3.32% and 3.02%, respectively. We consistently outperform CG-STVG significantly despite that CG-STVG mines relevant and beneficial instance context during decoding, which shows the great potential of our method. Because there are a lot of small objects and similar objects in the aerial scene, it is very difficult to mine discriminative instance context accurately by using single high-level scale. If the learned context contains irrelevant or even harmful information, the object queries will localize the wrong region in the decoding stage.

SGFDN can enjoy advanced performance with less computational complexity in natural video datasets, but has the worst performance in the SAVG task. This is precisely because its 3D decomposition strategy will lose a lot of object-related semantic information and motion information in aerial video, resulting in performance damage. This proves the challenge of the UAV-SVG dataset.

The suboptimal performance achieved by STCAT is attributed to the spatio-temporal consistency-aware transformer framework. The movement of UAVs, coupled with the motion of ground objects, causes the grounding model to face the drawbacks of feature alignment and grounding inconsistency. STCAT proposes global context modeling to generate multi-modal templates, enjoying more consistent cross-modal feature alignment and

grounding capabilities across frames. Our model still outperforms it on the m_vIoU and m_fAcc metrics with absolute gains of up to 3.01% and 1.65%, respectively.

Similar to VideoGrounding-DINO, TubeDETR adopts a DETR-like architecture enhanced by temporal aggregation modules. However, TubeDETR performs spatial multi-modal interactions over sparsely sampled frames. The difference between adjacent frames of aerial video is large and sparse sampling results in the loss of fine-grained spatio-temporal information.

In summary, the challenging SAVG task requires more delicate architectures or targeted improvements, and relatively generic improvements cannot significantly boost performance.

### D.3 Additional Ablation Studies

In this section, we conduct ablation studies to demonstrate the effectiveness of each component in our proposed SAVG-DETR framework.

#### D.3.1 Effect of the intra-scale multi-modality interaction branch

The intra-scale multi-modality interaction branch must perform local spatial modeling for each frame, followed by global temporal modeling across the entire aerial video. This is crucial for conducting multi-modality spatio-temporal interaction and modeling. To demonstrate the validity of this branch, we remove the temporal interaction layer (TIL) as a variant of our model. For two specific variants, video tokens are the average pooling or max pooling frame tokens, respectively. The quantitative ablation results are reported in Table 5 (a), which show a distinct performance drop without TIL. "SIL" represents the spatial interaction layer. In contrast, learnable tokens tend to be more equitable and flexible. In the TIL, they adaptively aggregate the information in frame tokens to model the global object semantic information. While other variants are generated directly from frame tokens, which involves biases to the specific context of the corresponding local spatial information.

#### D.3.2 Effect of the object queries initialization strategies

In Table 5 (b), we present ablation studies on the effectiveness of different object query initialization strategies. The initialization strategies in existing grounding methods mainly include learnable zero vector [18], average pooling or max pooling language tokens [1]. Among advanced visual grounding or video grounding methods, initializing object queries with zero vectors is the most common. As shown in the first row of Table 5 (b), this approach achieves similar performance to our method. When the object queries are initialized with the language tokens, performance degradation occurs. Specifically, average pooling sets the same attention weights equally for each word, max pooling selects one word token for each sentence to set its weight as 1 and that of others as 0. As empirically shown in the table, the best choice is to use video tokens for object query generation in our proposed query and position generator.

#### D.3.3 Ablation study on the multi-level progressive spatial decoder

The design choice of our proposed multi-level progressive spatial decoder is ablated on five aspects.

The time encoding and temporal self-attention are responsible for modeling the long-range temporal interactions in the object queries. As shown in Table 5 (c), our full decoder model is compared to variants without time encoding, without temporal self-attention, and without both. The variant without both is equivalent to a pure space-only decoder in the visual grounding task, which predicts each frame independently. The comparison shows that having temporal self-attention results in better performance. When using both time encoding and temporal self-attention, substantial performance gains are achieved over the space-only decoder.

As shown in Table 5 (d), our decoder is compared to variants without multi-level visual features, without multi-level language features, and without both. The variant without both corresponds to a decoder with a single high-level scale feature and cannot achieve competitive performance on the SAVG task. Language cross-attention and visual cross-attention enable queries to probe features within frames. The queries are enriched to produce the final contextualized representation used to generate the spatial tubes. Therefore, multi-level language or visual features can bring additional improvements (row 1 vs. row 2 or row 3). However, multi-level visual features can provide richer visual context for queries and gain more than multi-level language features (row 2 vs. row 3). Finally, the best performance can be achieved when using multi-level vision-language features. The language features modulated by multi-scale visual features can be used as a guide to help queries aggregate more object-related spatial information in multi-level visual features. Therefore, our full-fledged model is a great improvement over the single-scale decoder only (row 1 vs. row 4).

To demonstrate the effectiveness of our proposed progressive decoding paradigm based on offset generation, our decoder is compared to variants without the offsets generator, without positional embedding based on reference anchors, and without both. The variant without both degenerates into a framework with learnable positional embeddings and a cascade of decoder and prediction head. This degenerate baseline directly adopts the final

Table 7: Ablation on the scaling-up ratio of auxiliary loss.

| Methods | m_vIoU | vIoU@0.3 | vIoU@0.5 | m_fAcc | fAcc@0.3 | fAcc@0.5 |
|---|---|---|---|---|---|---|
| ($r$=0.90) | 19.60 | 27.74 | 16.36 | 20.61 | 21.36 | 16.33 |
| GIoU ($r$=1.00) | 26.55 | 37.84 | 22.20 | 28.15 | 34.20 | 25.42 |
| AuxGIoU ($r$=1.10) | 26.83 | 36.84 | **22.92** | 28.80 | 34.72 | 26.00 |
| AuxGIoU ($r$=1.15) | **27.15** | **38.18** | 22.85 | **28.82** | **35.85** | **26.55** |
| AuxGIoU ($r$=1.20) | 21.83 | 29.95 | 17.12 | 21.17 | 23.84 | 18.87 |

object queries to predict the spatial tubes, which is the classic paradigm among the advanced visual grounding or video grounding methods. As shown in Table 5 (e), our complete framework is a significant improvement over the baseline (row 1 vs. row 4). Furthermore, we can observe performance degradation in the other two variants (row 2 and row 3). Without positional embedding based on reference anchors, the offset generator cannot accurately predict the object's spatial offsets without the restriction of the grounding region (row 2). In contrast, object queries cannot accurately predict the object's spatial position in one step under the grounding region limitation (row 3). Our framework uses reference anchors to guide object queries at each decoding stage to predict offsets of the referred object and updates the reference anchor boxes for the next decoding stage. The framework iteratively refines reference anchors of the referred object, which helps to improve the prediction consistency.

In Table 5 (f), we evaluate our decoder when employing different numbers of decoder stages. As more decoder layers are used, the accuracy increases steadily until the saturation point is $M = 6$. This reflects the importance of multi-stage progressive reasoning for aerial video grounding. The progressive spatial decoder queries multi-level language features and collects multi-level visual information in multiple rounds, enabling the referred aerial object to be identified and localized more accurately. Since the accuracy does not improve at $M = 7$, we employ 6 decoder layers for our decoder by default.

Furthermore, we provide different level decoder combinations based on 6 decoder layers and decoding from high to low. The results are reported in Table 5 (g). We observe that under our decoding framework, the addition of lower-level details can gradually improve grounding performance. When the number of high, middle, and low-level decoders is 2, the second best performance can be achieved. When the number of low-level decoders is greater than 3, the high-level semantic information is reduced or even no, and the performance is significantly reduced (row 5 and row 6). This shows that our decoder achieves a better trade-off between low-level detail information and high-level semantic information.

### D.3.4 Effect of the scaling-up ratio of loss

To explore the effect of the scaling-up ratio, we start with 1 and gradually increase the ratio. In Table 7, our experiments demonstrate that setting the scaling-up ratio $r$ to 1.15 is an appropriate choice. When $r$ is 1, the auxiliary bounding box is the same as the actual bounding box, and the $\mathcal{L}_{AuxGIoU}$ degenerates to the $\mathcal{L}_{GIoU}$. When the ratio increases to 1.2, performance decreases significantly. This is due to the scale difference between the larger auxiliary bounding boxes and the actual bounding boxes. If the ratio is too large, the auxiliary bounding boxes cannot reflect the actual bounding box distribution and the quality of regression results, and the performance will be greatly lost. To prove that calculating IoU with the smaller scale auxiliary bounding boxes can not be beneficial for the presence of a large number of small object samples, we also set $r$ to 0.9. The results show that the smaller scale auxiliary bounding boxes will make the performance loss more serious.

### D.4 Visualization Analysis

### D.4.1 Qualitative Results

In Figure 16, we present more qualitative examples obtained from the UAV-SVG benchmark, comparing our results with other methods. In **Expressions 1**, **2**, and **3**, multiple small, similar moving objects of the same class appear in video frames, presenting a significant challenge. In **Expression 3**, most of the other methods localize other people playing basketball on the court. STCAT (yellow) localizes near the boy early on, but later experiences bounding box drift and is confused with the spectator outside the field. This is due to the semantic bias "on the right half of the court" that occurs later with the movement of the drone. However, our SAVG-DETR (red) provides stable tracking based on early results with high prediction consistency across frames. In the challenging example with low illumination in **Expression 4**, although some methods (STCAT (yellow), SGFDN (cyan), CG-STVG (pink)) can detect the referred object later, whereas our model maintains decent performance throughout the video. In **Expression 5**, it is necessary to understand both the object semantic and the complex spatial motion, and to reason about the regional location of the container truck. In Figures 18 and 19, we present more qualitative examples.

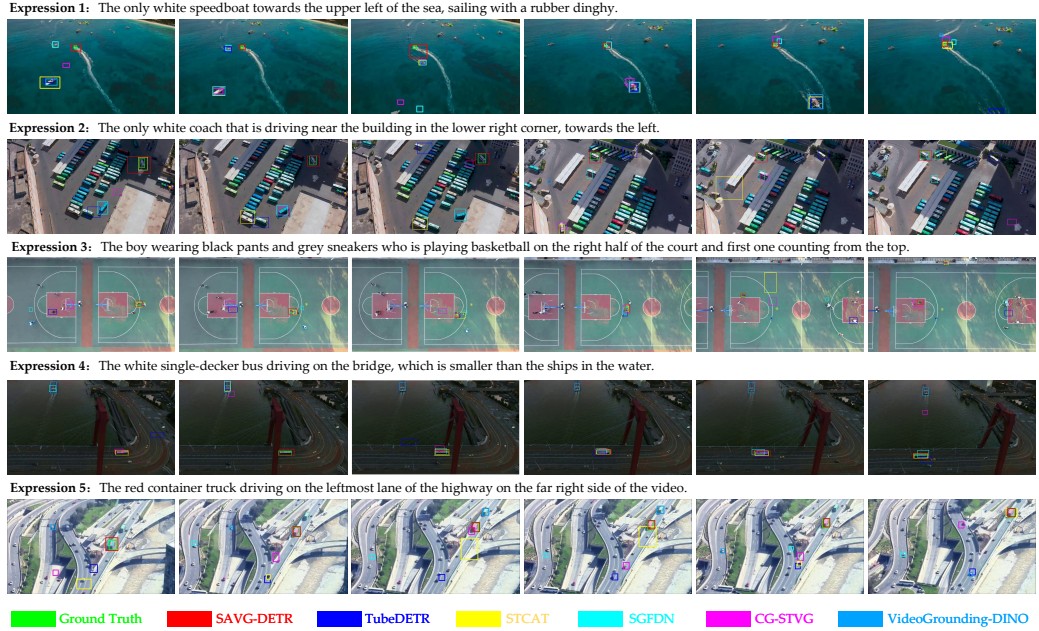

Figure 16: Qualitative results of different methods on the UAV-SVG benchmark. Best viewed by zooming in.

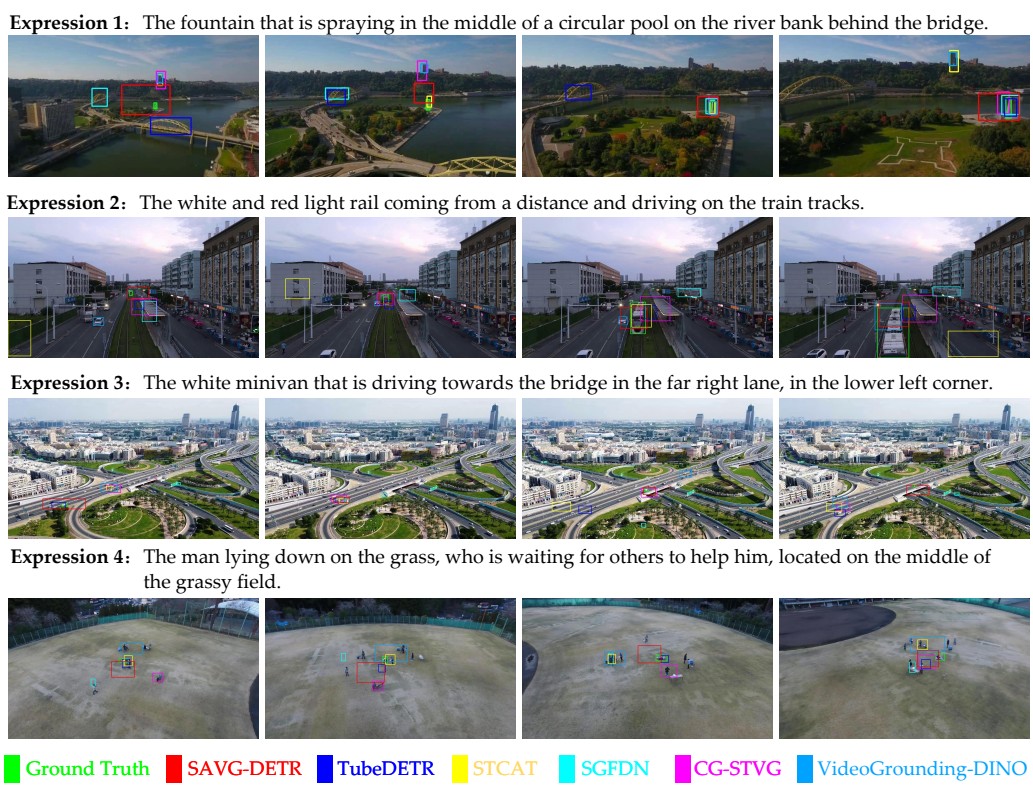

Figure 17: Some failure qualitative examples of the proposed method. Best viewed by zooming in.

### D.4.2 Failure Analysis

In Figure 17, we show some failure examples. In **Expression 1**, as the drone gets closer to the fountain, some methods (SGFDN (cyan), CG-STVG (pink), TubeDETR (blue)) gradually localize the object. Although our SAVG-DETR (red) fails to locate it accurately, by reasoning with the spatial position of the circular pool-river bank-bridge, the red box can cover the green object region. In **Expression 2**, all methods struggle to locate the "light rail" or only partially detect the object. Our SAVG-DETR (red) can cover its semantic region when the light rail is far away. In **Expression 3**, the video showcases an extremely tiny white minivan, and there are many similar small white vehicles. In this challenging example, all methods fail to locate it, whereas our model predicts a large enveloping box (red) for stable tracking of the small "minivan" (green). Other methods often predict semantically irrelevant regions. In **Expression 4**, this scenario involves significant viewpoint changes. Our bounding box (red) can not accurately localize the region (green), but it is adjacent to the region of the "man lying down on the grass". Even in the case of a final grounding failure, our SAVG-DETR framework can maintain semantic relevance and stable consistency across frames. This is attributed to our decoding paradigm based on offset generation.

## E  Licenses

We choose WebUAV-3M [9] as our data source. We have properly credited the creators or original owners of assets used in the paper, and we follow the license GNU General Public License v3.0.

## F  Limitations and Future Work

Although our method is specifically designed for the SAVG task and surpasses the existing SoTA methods, achieving the best balance between performance and complexity at the same time, there are still some limitations. First, our method does not fully explore the real-time performance in practical applications. In practical applications, real-time grounding and tracking of the object are required. Second, aerial scenes often have occlusion situations, and our method has not been optimized for this problem yet.

In the future, we consider further expanding the multi-modality and multi-scale spatio-temporal modeling method to further enhance the model's ability to capture small-scale objects and occlusion objects. Moreover, it is also necessary to explore the potential for smaller sizes and faster models.

## G  Societal Impact

In light of the fact that the era of low-altitude economy and the field of spatial temporal intelligence has just begun, we establish a comprehensive benchmark for future advancements in spatial aerial video grounding. To the best of our knowledge, this is the first to support the video grounding in the low-altitude UAV. With the widespread application of unmanned aerial vehicles (UAVs) globally, many tasks are currently performed in the sky, such as UAV-based goods delivery, urban traffic/security patrol, industrial inspection, disaster rescue in bad weather, and scenery tour. Since SAVG can naturally combine UAVs' visual and text signals to complete object localization and tracking more effectively, achieving this grounding ability is crucial for advancing towards low-altitude intelligence. We believe this work will open up avenues for this new kind of SAVG.

However, this also raises concerns about how the SAVG model with strong tracking capabilities could be inappropriately used in the community, such as for illegal UAV surveillance and tracking.

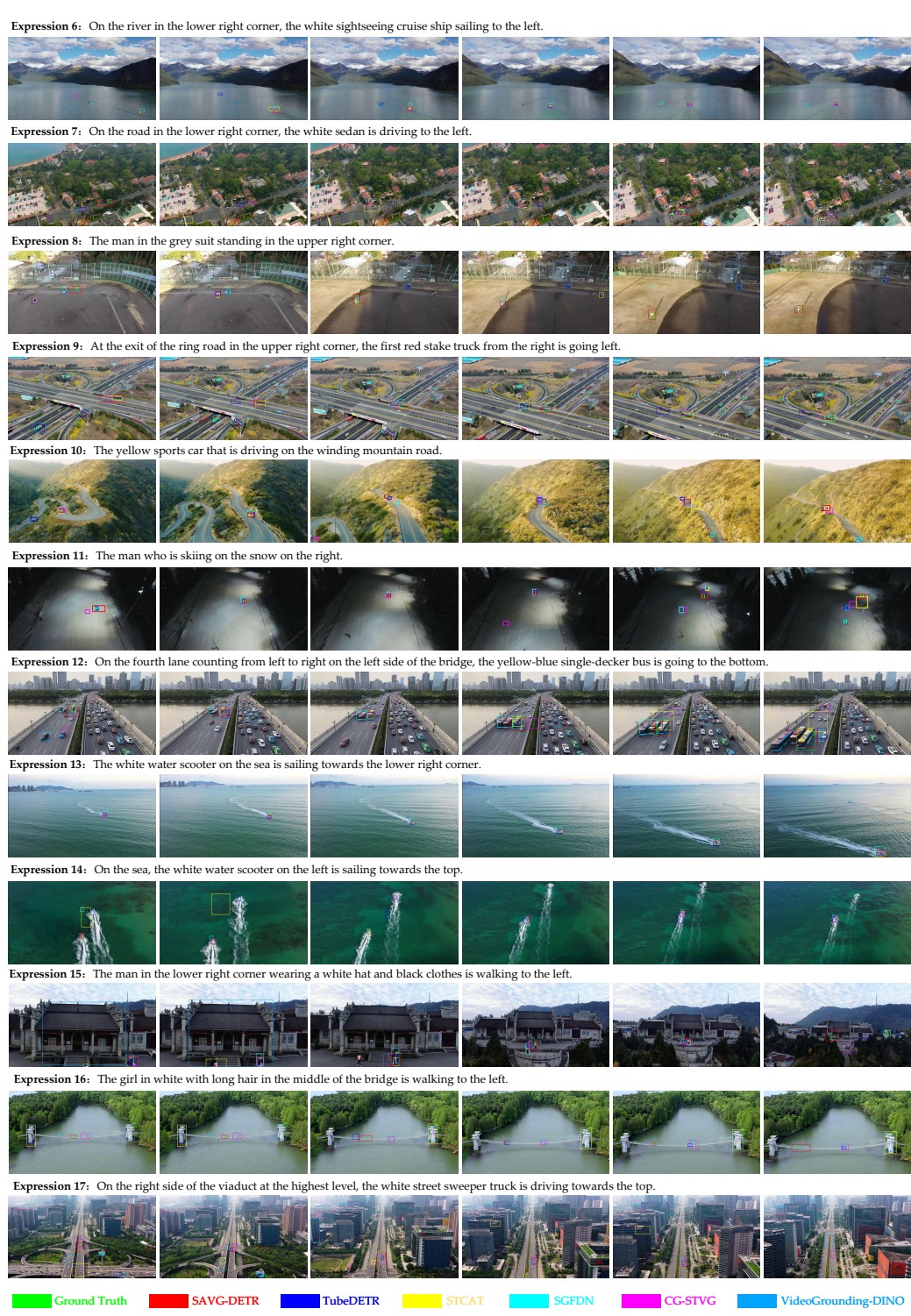

Figure 18: More qualitative results. Continuation of Figure 16.

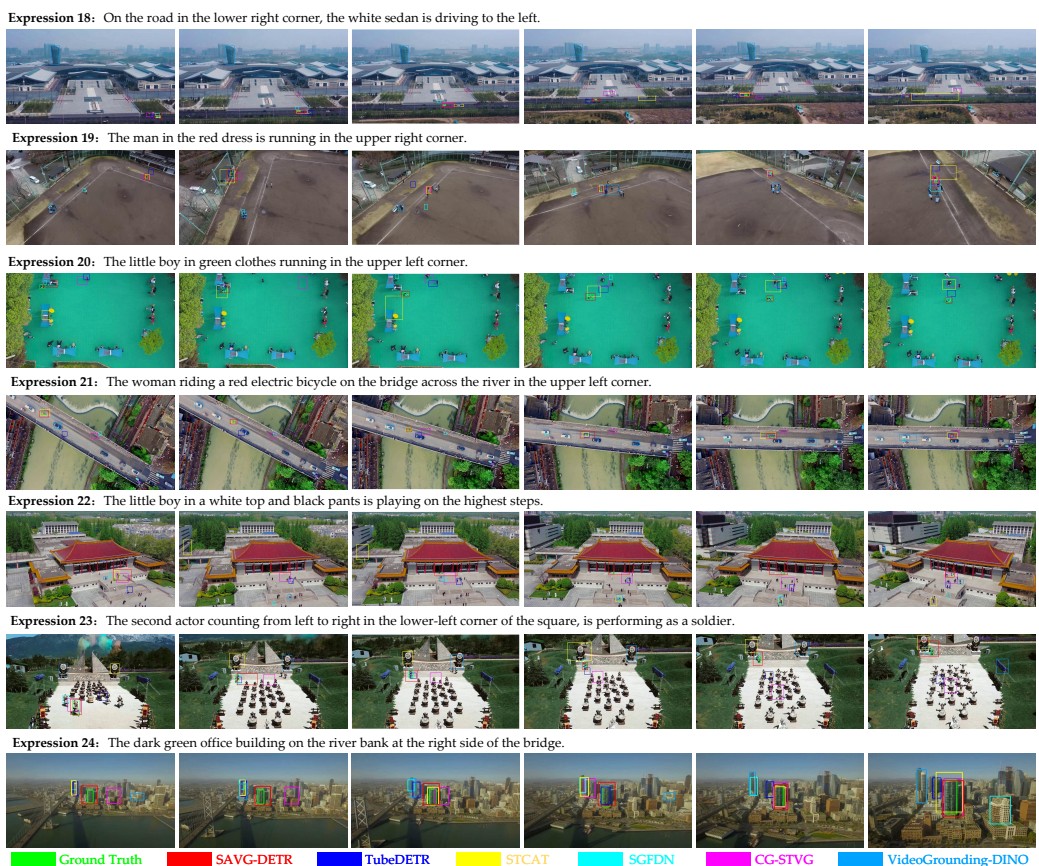

Figure 19: More qualitative results. Continuation of Figure 18.

