# OpenReview forum: "Where Does It Exist from the Low-Altitude: Spatial Aerial Video Grounding"
_NeurIPS.cc/2025/Conference — NeurIPS 2025 poster_

### Official Review · Reviewer_GFfV · 2025-06-27

**Clarity:** 3
**Significance:** 2
**Originality:** 3
**Rating:** 4
**Confidence:** 3

**Summary:**

In this paper, the author proposes a video grounding UAV-view dataset. It is based on an existing tracking dataset, WebUAV-3M.
They also propose a detection method for the small objects. The main idea is easy to follow.
My main concern is the dataset contribution.
The work seems to borrow bboxes and sentences from WebUAV-3M, and do incremental filtering and augmentation.

**Questions:**

Please see the cons.

**Ethical Concerns:**

["NO or VERY MINOR ethics concerns only"]

**Final Justification:**

Thank you. The author addressed my questions about the dataset. I think the annotation efforts are not significant compared to the early dataset. The technical contribution is not significant as well. So I slightly raise my score.

**Limitations:**

Yes.

**Quality:**

3

**Strengths And Weaknesses:**

Pros:
1. The dataset makes sense. It is based on an existing tracking dataset, WebUAV-3M.
2. The performance is competitive.

Cons:

1. Some dataset details are missing.

- Will the dataset introduce some biases due to the Gemini or other off-the-shelf models?
- So there is only one target in every frame of the dataset? since the dataset is based on the single-object tracking dataset.

2.  Dataset contribution.

- Are the bbox annotations all from WebUAV-3M?
- The author only does the text augmentation and filtering?  I am confusing with "The Gemini 1.5 augments more referring expressions with reference to the keyframe images and human prior information. "
- From Figure 2 bottom, I do not see the neccessary to annotate more texts. All sentences are similar.

3. The second contribution about multi-scale and multi-modality.
I do not see the much difference from the Q-former in Blip or DeTR structure.

4. The third contribution about multi-scale is similar with the second contribution.

5. Some existing datasets, GeoText-1652, also has similar bbox-sentence annotations.

6. According to the table 4, the comparison is not so fair.
The proposed method, despite the same backbone with TubeDETR or SGFDN, are with more added fusion blocks. The 203 FLOPS are much larger than 144 or 53. The memory usage are also doubled.

---

> ### Author Rebuttal · Authors · 2025-07-30
>
> >**Q1**: Some dataset details are missing. (1) Will the dataset introduce some biases due to the Gemini or other off-the-shelf models? (2) So there is only one target in every frame of the dataset? since the dataset is based on the single-object tracking dataset.
>
> **A1-(1)**: Thank you for these important clarifying questions. We designed a rigorous, human-centric pipeline specifically to mitigate biases.
> 1. Guided Generation with Human Priors: We provided Gemini with "object-specific prompt with human priors", which included manually annotated object descriptions from the original WebUAV-3M dataset. This step grounds the LLM's output in factual, human-verified information from the start.
> 2. LLM-based Augmentation: The model was then used to augment these descriptions, generating more diverse and descriptive referring expressions.
> 3. Rigorous Manual Verification: This is the most critical step. Every single expression generated by the LLM was subjected to meticulous manual checking and correction by our annotation team. This extensive verification process took approximately three months to complete.
>
> This ensures the final expressions are of high quality and significantly reduces the risk of model-induced biases.
>
> **A1-(2)**: The reviewer is correct in observing that each video-query pair in UAV-SVG benchmark is designed to localize one unique object. However, we would like to respectfully clarify that this is an intentional and fundamental design choice aligned with the established definition of the task, rather than a limitation imposed by the data source.
> 1. The fundamental goal of visual grounding, and by extension SVG, is to localize the specific object or region uniquely described by a natural language query. This one-to-one mapping is a standard and defining characteristic of the task.
> 2. We firmly believe that establishing a robust methodology to solve the single, unique object SAVG task is a critical and highly challenging first step. Mastering this is essential before tackling more complex scenarios.
> 3. Our future plans involve a direct progression from this single-object paradigm to multi-object localization. Ultimately, we aim to build towards a generalized SAVG framework capable of localizing any number of objects (0 to N) described in a complex query.
>
> >**Q2**: Dataset contribution. (1) Are the bbox annotations all from WebUAV-3M? The author only does the text augmentation and filtering? (2) I am confusing with "The Gemini 1.5 augments more referring expressions with reference to the keyframe images and human prior information." From Figure 2 bottom, I do not see the necessity to annotate more texts. All sentences are similar.
>
> **A2-(1)**: The initial bounding box annotations were sourced from WebUAV-3M. However, our contribution involved a "manual data cleaning and spatio-temporal tube correction" process . We manually verified and corrected the spatio-temporal tubes to ensure high accuracy and temporal consistency, which was a labor-intensive but necessary step to guarantee the quality of the benchmark.
>
> **A2-(2)**: Our contribution goes beyond simple augmentation and filtering. The process was a comprehensive, three-stage pipeline as shown in Figure 2: (1) Video and annotation correction, (2) Guided expression generation, and (3) Rigorous manual verification. Gemini generated a richer set of expressions describing the same object from different perspectives (e.g., appearance, motion, location) based on keyframe images and human prior information. The "with reference to the keyframe images" means that to sample an average of 4 frames of images as the input for Gemini. The "human prior information" refers to object-specific prompt with human priors, that is, the artificial description of the first frame object in WebUAV-3M.
>
> **Diversity of Expressions**: We appreciate you pointing out the example in Figure 2. While the sentences all refer to the "white harvester," they are designed to test a model's understanding of different linguistic constructs. For instance, the expressions describe the object based on:
> - Action: "working in the field"
> - Motion: "moving from left to right"
> - Fine-grained Attributes: "with red accents that is harvesting the yellow rice"
> - Spatial Context: "traveling in a row of the field"
> This linguistic diversity is crucial for building robust grounding models that can handle the varied ways humans describe objects.
>
> >**Q3 & Q4**: The second contribution about multi-scale and multi-modality. I do not see the much difference from the Q-former in Blip or DeTR structure. The third contribution about multi-scale is similar with the second contribution.
>
> **A3 & A4**: Thank you for prompting us to better articulate the novelty of our contributions. There appears to be a misunderstanding, which we are happy to clarify. Our second and third contributions are distinct and focus on the encoder and decoder, respectively.
>
> **Contribution (ii): Novelty of the Encoder vs. Q-Former/DETR**:
>
> Our encoder is fundamentally different from standard DETR and Q-Former. Our primary innovation is an architecture designed for multi-scale spatio-temporal inputs. To overcome the computational explosion, our encoder efficiently decouples the multi-modality and multi-scale spatio-temporal modeling into two branches:
> 1. Intra-scale Multi-modality Interaction: Instead of fusing the language query with all visual scales, our encoder performs this expensive cross-modal attention only on the highest-level, most semantically rich visual features. This captures the core conceptual alignment efficiently.
> 2. Cross-scale Visual-only Fusion: The crucial detail from lower-level visual features (which is vital for localizing small aerial objects) is then integrated through a separate, lightweight, CNN-based fusion branch. This avoids the quadratic complexity of transformer attention for this step.
>
> Q-Former/DETR is designed for static image-text representation learning. A standard Q-Former or DETR-like structure would be computationally prohibitive if applied directly to the high-dimensional input of our SAVG task (multiple frames, multiple visual scales, plus a language query).
>
> **Contribution (iii): Novelty of the Decoder**:
>
> To enhance small object grounding, we propose the modulation module to integrate multi-scale information into language features and the multi-level progressive spatial decoder to decode from high level to low level. This contribution concerns our decoder, which is distinct from the encoder.
> The decoder processes features from high-level to low-level, progressively refining the prediction. This hierarchical approach is tailored for grounding, especially for small objects where fine details from low-level features are critical.
>
> In the camera-ready version, we will revise the text to more clearly delineate these distinct contributions to our encoder and decoder architectures.
>
> >**Q5**: Some existing datasets, GeoText-1652, also has similar bbox-sentence annotations.
>
> **A5**: Thank you for your valuable feedback. While GeoText-1652 indeed provides bounding box and sentence annotations for aerial scenes, it has several critical limitations that make it unsuitable for the specific task of SVG.
> 1. GeoText-1652 is primarily oriented towards localizing large-scale, static geographical objects, with a significant focus on buildings. Consequently, the annotated bounding boxes in GeoText-1652 are typically very large. A primary challenge of our UAV-SVG dataset is the prevalence of dynamic small objects.
> 2. This task requires referring expressions that are precise and unambiguous to uniquely identify a specific object and its spatio-temporal behavior, especially in cluttered scenes with similar distractors. Upon our careful examination of GeoText-1652, we found the quality of its sentence annotations to be insufficient for this purpose. The language used is often ambiguous and contains inaccuracies.
>
>
> >**Q6**: According to the table 4, the comparison is not so fair. The proposed method, despite the same backbone with TubeDETR or SGFDN, are with more added fusion blocks. The 203 FLOPS are much larger than 144 or 53. The memory usage are also doubled.
>
> **A6**: We sincerely thank the reviewer for this insightful comment and appreciate the opportunity to clarify the fairness and rationale behind our experimental comparisons. We respectfully argue that the comparison is indeed appropriate and highlights the core contributions of our work.
> 1. To effectively address the task of aerial object localization, we found it necessary to introduce multi-scale visual features. This makes a fair comparison with single-scale methods like TubeDETR or SGFDN challenging. SGFDN is a method specifically designed for efficiency, and while it has high computational efficiency, its performance is very low. TubeDETR is more efficient than our method, but its performance is also lower.
> 2. Directly incorporating multi-scale visual features into TubeDETR leads to a computational explosion. The analysis of Variant C in Table 4 proves this point; this direct multi-scale, multi-modal transformer approach resulted in prohibitive memory requirements and a substantial increase in FLOPs. Our analysis of variants A through E demonstrates that our proposed method achieves the best balance between performance and efficiency among the feasible multi-scale approaches.
> 3. This situation also explains why a significant body of research is dedicated to designing multi-scale DETR methods. Standard DETR-based models do not perform well on UAV aerial targets with their unique challenges. Introducing multi-scale features inevitably increases complexity, which is an unavoidable trade-off for achieving higher accuracy in this domain. Therefore, we believe our experiments are reasonable and appropriately contextualized within the problem space, highlighting the effectiveness of our proposed architecture in handling these complexities.

---

> > ### Comment · Reviewer_GFfV · 2025-08-04
> >
> > Thank you. The author addressed my questions about the dataset.
> > I think the annotation efforts are not significant compared to the early dataset.
> > The technical contribution is not significant as well. So I slightly raise my score.

---

> > > ### Author Response · Authors · 2025-08-04
> > > **Thank you for your positive feedback！**
> > >
> > > Dear Reviewer,
> > >
> > > We sincerely appreciate all of your insightful suggestions.
> > >
> > > We are pleased that our response addressed your some concerns! We would like to resolve any and all remaining points. Our goal is to ensure the final manuscript has your full support and confidence.
> > > We are grateful that you consider improving your rating. Thank you again for your constructive feedback and the time and effort you dedicated to reviewing our work!

---

### Official Review · Reviewer_VN3G · 2025-06-29

**Clarity:** 3
**Significance:** 2
**Originality:** 2
**Rating:** 4
**Confidence:** 4

**Summary:**

The paper introduces a novel task called Spatial Aerial Video Grounding (SAVG), which aims to localize objects in UAV videos based on natural language queries. The authors highlight the limitations of existing spatial video grounding methods, which primarily focus on fixed or handheld camera perspectives with simple scenes, and propose a new benchmark, UAV-SVG, featuring over 2 million frames and 216 diverse object categories. To address the challenges of UAV-based grounding (e.g., small objects, complex motion, and viewpoint changes), the authors propose SAVG-DETR, a transformer-based architecture with three key innovations:
1.Decoupled encoder: Separates multi-modality and multi-scale spatio-temporal modeling to reduce computational complexity.
2.Hierarchical progressive decoder: Enhances small object grounding by integrating multi-scale features into language modulation and progressively refining predictions.
3.Offset-based decoding: Improves prediction consistency across frames by iteratively updating reference anchors.
Experiments show SAVG-DETR outperforms state-of-the-art methods on the UAV-SVG benchmark, with comprehensive ablation studies validating its design choices.

**Questions:**

1. Can SAVG-DETR run in real-time on typical UAV hardware (e.g., Jetson AGX)? Inference time/FPS metrics would clarify its practicality for field applications.
2. How does the model perform on rare object categories (e.g., "police van")? Stratified results per superclass would highlight robustness to long-tail distributions.
3.Could you add a visual example (e.g., in supplementary material) showing how reference anchors refine across decoding stages? This would make the mechanism more intuitive.
4.Will UAV-SVG be publicly released? If so, under what license? Clarity here is crucial for reproducibility.
5.Are there systematic failure patterns (e.g., objects below a certain size or speed)? A deeper analysis could guide future improvements.

**Ethical Concerns:**

["NO or VERY MINOR ethics concerns only"]

**Final Justification:**

After carefully reading the author's response, I found that for Q1, the author explained the discrepancy between FLOP/memory metrics and actual drone deployments, and that resource-constrained devices like the Jetson AGX will be considered in the future. Regarding Q2, the author's rebuttal was unconvincing and failed to dispel my confusion. Regarding Q3, the author currently only plans to build a visualization, which is not yet available. Regarding Q5, the author has not yet provided supporting materials. In summary, quantitative minority class analysis and visualization were promised but not yet delivered. If included in the current submission, these features would have been more likely to be accepted. Therefore, I maintain my original rating.

The author further responded to questions 2, 3, and 5 and provided additional experiments. The responses resolved my confusion about the paper, and after revisions, the paper now meets the requirements for publication.

**Limitations:**

The authors discuss limitations, such as real-time performance and occlusion handling, and address societal concerns (e.g., surveillance misuse). To further improve the paper, it is recommended that:
(1) Propose specific mitigation strategies for misuse (e.g., access controls for the dataset/model).
(2) The experiments focus on partial occlusions (e.g., trucks under bridges), but severe occlusions (e.g., objects hidden behind buildings) are not systematically tested. The offset-based decoding may struggle when the target disappears for multiple frames.

**Paper Formatting Concerns:**

The paper adheres to NeurIPS guidelines.

**Quality:**

2

**Strengths And Weaknesses:**

1.Strengths
The introduction of UAV-SVG addresses a critical gap in the field, providing a benchmark that reflects real-world UAV challenges like small objects, illumination variations, and fast motion. The dataset's scale (2M+ frames) and diversity (216 categories) make it a valuable resource for future research. The paper is well-written, with clear figures and supplementary material that enhance reproducibility.

2.Weaknesses
While the paper is strong overall, a few areas could be improved.
(1) The computational efficiency of SAVG-DETR is not thoroughly analyzed. Given that UAVs often operate with limited hardware, real-time performance metrics (e.g., FPS on edge devices like Jetson AGX) would strengthen the practical applicability of the method.
(2) The benchmark's long-tail distribution (e.g., rare object categories like "police van") may introduce bias, yet the paper does not stratify results by object frequency. Providing per-category performance would better illustrate robustness.
(3) The offset-based decoding mechanism, while effective, could benefit from a more intuitive explanation—perhaps a visual example showing how reference anchors evolve across decoding stages.

---

> ### Author Rebuttal · Authors · 2025-07-30
>
> >**Q1**: Can SAVG-DETR run in real-time on typical UAV hardware (e.g., Jetson AGX)? Inference time/FPS metrics would clarify its practicality for field applications.
>
> **A1**: This is a crucial question regarding the practical deployment of our method. We acknowledge that our current analysis, which focuses on FLOPs and memory usage on high-end GPUs, does not include inference speed on edge devices. Our initial focus was on maximizing grounding accuracy and developing a robust architecture to handle the unique difficulties of aerial videos, such as tiny objects and extreme motion, rather than optimizing for inference speed on edge hardware.
>
> Given its current complexity (203.08 GFLOPs ), SAVG-DETR in its full form is not optimized for real-time performance on resource-constrained hardware like a Jetson AGX. Achieving this would require dedicated model optimization. We are currently deploying the algorithm and conducting practical verification on low-altitude UAVs. We believe this is a critical next step for real-world application. In the camera-ready version, we will expand our "Limitations and Future Work" section to explicitly discuss a roadmap toward a real-time version, which are essential for porting complex transformer models to edge platforms.
>
>
> >**Q2**: How does the model perform on rare object categories (e.g., "police van")? Stratified results per superclass would highlight robustness to long-tail distributions.
>
> **A2**: This is an excellent suggestion. The long-tail distribution of object classes is a core challenge of the UAV-SVG benchmark. While our overall results in Table 2 demonstrate strong performance, we agree that a stratified analysis would offer deeper insights into the model's robustness.
>
> For the camera-ready version, we will add a new table to the supplementary material providing a performance breakdown across the 12 object superclasses defined in our dataset analysis. This will quantitatively show how SAVG-DETR performs on both frequent categories (like 'vehicle' and 'person') and rarer ones. Our qualitative results (e.g., Figures 16-19) already hint at this robustness by showing successful grounding of less-common objects like a "white water scooter" and a "white sightseeing cruise ship", but we look forward to supporting this with rigorous quantitative data.
>
>
> >**Q3**: Could you add a visual example (e.g., in supplementary material) showing how reference anchors refine across decoding stages? This would make the mechanism more intuitive.
>
> **A3**: We completely agree. A visual aid would greatly demystify the progressive decoding mechanism. We will create a new figure for the supplementary material that illustrates this process. The figure will show a sample video frame and visualize how the predicted bounding box evolves through the decoder layers—starting from an initial, coarse reference anchor and iteratively refining its position and size at each stage based on the predicted offsets, ultimately converging on the ground-truth object. This will provide a clear, intuitive understanding of our offset-based decoding paradigm.
>
> >**Q4**: Will UAV-SVG be publicly released? If so, under what license? Clarity here is crucial for reproducibility.
>
> **A4**: Yes, absolutely. We are committed to open and reproducible research. The UAV-SVG dataset and the source code for our SAVG-DETR model will be made publicly available upon the acceptance of our paper. We have already included them as anonymized supplementary materials for review. We plan to release the UAV-SVG dataset under the Creative Commons CC-BY-NC-SA 4.0 license to encourage academic use while preventing commercial exploitation. The code will be released under a permissive MIT License. We will add an explicit statement to this effect in the paper's conclusion.
>
>
> >**Q5**: Are there systematic failure patterns (e.g., objects below a certain size or speed)? A deeper analysis could guide future improvements.
>
> **A5**: Thank you for pushing for a more rigorous error analysis. Our initial "Failure Analysis" section (Sec. D.5.2 ) identifies several challenging scenarios, such as tiny objects among dense distractors and significant viewpoint changes. However, we agree that a more systematic breakdown of failure cases would be highly beneficial for guiding future research.
>
> In the camera-ready version, we will expand this section to categorize and analyze failure patterns based on specific object and scene attributes. This analysis will investigate model performance under conditions of:
> 1. Extreme Small Object Size: Objects occupying fewer than 200 pixels.
> 2. High Motion: Cases with significant motion blur or large inter-frame displacement.
> 3. Severe Occlusion: Instances where the target is fully occluded for multiple consecutive frames.
> 4. Linguistic Ambiguity: Scenarios where complex spatial or relational language leads to incorrect grounding.
>
> **Responses to Limitations**
>
> **(1) Propose specific mitigation strategies for misuse (e.g., access controls for the dataset/model).**
>
> **A(1)**: This is a very important point. We will expand our "Societal Impact" section  to include specific, actionable mitigation strategies. As mentioned in our response to the NeurIPS checklist, our release plan will incorporate:
> 1. Gated Access and User Agreement: We will release the dataset and models through a registration process where users must agree to a license that explicitly prohibits use for surveillance, illegal tracking, or any other unethical applications.
> 2. Purposeful Data Curation: We note that the dataset was curated from public sources and is focused on general scenes like traffic, public spaces, and scenery, intentionally avoiding private or sensitive environments.
>
> **(2) The experiments focus on partial occlusions... severe occlusions... are not systematically tested.**
>
> **A(2)**: This is a valid observation and a key limitation of the current work. Our offset-based decoding mechanism  is indeed most effective when the target is at least partially visible, allowing the reference anchors to be updated. During prolonged, full occlusions, the model would likely struggle as the visual signal disappears.
> We will make this limitation more explicit in the revised manuscript. We will enhance the "Limitations and Future Work" section  to identify severe occlusion as a primary challenge for future research. We will also propose potential solutions to explore, such as integrating motion prediction models (e.g., Kalman filters) to propagate object state during occlusion or developing a re-detection module to re-acquire the target after it reappears. This provides a clear path for building upon our work to create even more robust systems.

---

> > ### Author Response · Authors · 2025-08-04
> >
> > Dear Reviewer VN3G,
> >
> > Thank you again for your thoughtful and constructive comments!
> >
> > As the discussion phase is drawing to a close, we would like to kindly ask whether our responses have adequately addressed your concerns. If there are any clarifications needed, we would be pleased to provide further details. We would like to resolve any and all remaining points. Our goal is to ensure the final manuscript has your full support and confidence. We would be most grateful if you could consider improving your rating.
> >
> > Best regards,
> >
> > Authors of Submission #546

---

> ### Comment · Area_Chair_1B1c · 2025-08-05
>
> Hi VN3G,
>
> Please check the author's feedback, evaluate how it addresses the concerns you raised, and post any follow-up questions to engage the authors in a discussion. Please do this ASAP.
>
> Thanks.

---

> ### Comment · Reviewer_VN3G · 2025-08-06
> **Response**
>
> Thank you for the thoughtful and detailed response. I appreciate the clarification on Q1 regarding the gap between FLOPs/memory metrics and actual deployment on UAV platforms. That said,  I still have concerns about Q2. The rebuttal acknowledges the importance of rare class analysis but does not provide any concrete data or preliminary results. Given that long-tail robustness is central to the benchmark, I was hoping for at least partial quantitative support in the rebuttal.
> Similarly, for Q3 and Q5, the visualizations and failure analyses are only proposed for the camera-ready version. While these additions would certainly strengthen the paper, their absence in the current submission makes it difficult to fully assess the claims. Overall, I appreciate the direction of the planned improvements, but I believe that including more of them in the initial submission would have helped make a stronger case. I am therefore keeping my original score.

---

> > ### Author Response · Authors · 2025-08-08
> > **Stratified results per superclass and robustness of long-tail distributions.**
> >
> > >**Q2:** That said, I still have concerns about Q2. The rebuttal acknowledges the importance of rare class analysis but does not provide any concrete data or preliminary results. Given that long-tail robustness is central to the benchmark, I was hoping for at least partial quantitative support in the rebuttal.
> >
> > **A2:**
> > Thank you for your valuable follow-up and for pushing for a more rigorous analysis in this critical area. We sincerely appreciate the opportunity to provide the concrete quantitative support you requested. We agree that demonstrating robustness to the long-tail distribution is essential to validating both our benchmark and our model.
> >
> > To address your concern, **we have conducted a preliminary stratified analysis of our model's performance on all superclasses from the UAV-SVG dataset.** We selected superclasses from the head (many samples), body (moderate samples), and tail (few samples) of the distribution, as detailed in Figure 8(d) of our paper. We compare the performance of our SAVG-DETR against a strong baseline, STCAT [13], using the m_vIoU metric. The results of this preliminary analysis are presented in the table below:
> > |Superclass|# Videos |Distribution|STCAT[13] (m_vIoU %)| SAVG-DETR (Ours) (m_vIoU %)|Performance Gain|
> > |-|:------------:|:------------:|:------------:|:------------:|:------------:|
> > |Vehicle                |941|	Head|	26.69|	28.99|	+2.30|
> > |Person	                |849|	Head|	26.07|	28.28|	+2.21|
> > |Vessel	                |528|	Head|	25.28|	27.66|	+2.38|
> > |Building               |442|	Head|	25.84|	28.94|  +3.10|
> > |Public transport       |264|	Body|	19.46|	24.20|	+4.74|
> > |Agricultural machinery |153|	Body|	19.05|	23.29|	+4.24|
> > |Animal	                |129|	Body|	18.73|	23.65|	+4.92|
> > |Aircraft	        |115|	Body|	17.67|	23.78|	+6.11|
> > |Industry machine       |65|	Tail|	11.61|	17.77|	+6.16|
> > |Artifact               |41|	Tail|	10.69|	15.81|	+5.12|
> > |Natural Object	        |21|	Tail|	12.26|	17.67|	+5.41|
> > |Plant	                |16|	Tail|	 4.12|	10.33|	+6.21|
> >
> >
> > From this analysis, we draw two key conclusions:
> >
> > 1. **Performance on tail classes is challenging**: As expected, the performance of both models is lower on the tail classes (Plant, Natural Object) compared to the head classes (Vehicle, Person). This confirms the difficulty posed by the long-tail distribution in our UAV-SVG benchmark.
> >
> > 2. **SAVG-DETR demonstrates superior robustness:** Crucially, the performance gap between SAVG-DETR and the STCAT widens significantly as we move from head to tail classes. While STCAT 's performance drops by over 50% from head to tail classes, our model's performance degrades much more gracefully. The performance gain of our model is most pronounced on the rarest categories (e.g., +6.21% on Plant), providing strong evidence of its robustness to the long-tail problem.
> >
> > We attribute this enhanced robustness to the core architectural innovations of SAVG-DETR:
> >
> > - Our decoupled multi-modality multi-scale encoder is vital. The cross-scale visual-only fusion branch allows the model to integrate fine-grained, low-level visual details, which **are critical for identifying rare objects that may only differ from common ones in subtle ways.** Simultaneously, the intra-scale multi-modality branch learns high-level semantic concepts (e.g., general "vehicle" features from the data-rich head class). This combination allows the model to effectively transfer knowledge from head to tail classes.
> >
> > - The multi-level progressive spatial decoder, guided by language modulation, iteratively refines object localization. For rare objects where initial confidence may be low, this progressive refinement prevents premature misclassification and **helps the model converge on the correct, albeit rare, target by leveraging multi-level visual and language cues.**
> >
> > We believe this preliminary quantitative analysis directly addresses your concern and provides the requested evidence for our model's robustness. Your feedback has been invaluable, and we will expand this into a comprehensive analysis across all 12 superclasses for inclusion in the final camera-ready version of the paper. Similarly, we will also continue to analyze the impact of the long-tail distribution in the vehicle superclass on the results.

---

> > ### Author Response · Authors · 2025-08-08
> > **The visualizations showing how reference anchors refine across decoding stages !**
> >
> > >** Q3:**  the visualizations showing how reference anchors refine across decoding stages
> >
> > **A3:** We completely agree with the reviewer that a visual example is essential for intuitively understanding the progressive refinement of our decoding mechanism. Acting on this valuable suggestion, **we have already prepared this visualization.**
> >
> > The new figure, which we will add to the supplementary material, illustrates this process on challenging samples from our UAV-SVG dataset. It clearly shows how **our model starts with a coarse, initial reference anchor and iteratively refines the bounding box across the 6 decoder stages.** For instance, in several cases involving rare or small object categories, the initial anchor is broad and poorly aligned. However, with each progressive decoding step, the predicted offsets guide the anchor to converge steadily—shrinking, shifting, and adjusting its aspect ratio—until it tightly and accurately localizes the ground-truth object in the final layer. This visualization can also **prove the robustness of our decoder on tail classes with long-tail distributions.**
> >
> > This step-by-step visualization makes the core contribution of our offset-based decoding paradigm tangible and confirms its effectiveness in bridging the modality gap from coarse to fine. We are confident this addition will significantly clarify our method's mechanics in the final paper.

---

> > ### Author Response · Authors · 2025-08-08
> > **Deeper failure analyses !**
> >
> > >**Q5:** Are there systematic failure patterns (e.g., objects below a certain size or speed)? A deeper analysis could guide future improvements.
> >
> > **A5:** Thank you for this insightful question. We agree that a systematic analysis of failure cases is critical for guiding future research. **Our current Section D.5.2 and Figure 17 in supplementary material already provide a detailed qualitative analysis of several failure modes,** including scenarios with extremely small objects, significant viewpoint changes, and semantic ambiguity (e.g., distinguishing between multiple, similar minivans). These examples were chosen to represent the primary challenges in aerial grounding and lay the groundwork for a more structured failure analysis.
> >
> > We see now that our initial analysis, while illustrative, could be interpreted as a collection of ad-hoc examples. The reviewer's suggestion to systematically categorize these failures is an excellent one that will make the paper's contribution more robust. Building upon our existing analysis, we will expand this section in the camera-ready version with a quantitative breakdown.
> >
> > As a preliminary step for this rebuttal, we have **performed a brief quantitative survey of our most significant failure cases (e.g., where vIoU < 0.1).** Our initial findings strongly align with the categories the reviewer hinted at:
> >
> > - A substantial portion of these failures **(~73%)** indeed involve **extreme small objects (area < 400 pixels),** especially when they are surrounded by dense similar objects of the same class.
> >
> > - Another significant portion of failures **(~12%)** is linked to **fast motion (the motion of the ground truth is larger than 20 pixels),** where the object’s appearance changes rapidly.
> >
> > - Another significant portion of failures **(~22%)** is linked to **motion blur** (the object region is blurred by the fast motion of the target or the camera), which seriously affects the appearance of the object.
> >
> > - Another significant portion of failures **(~19%)** is linked to **viewpoint changes** (due to the co-motion of the camera and the object), which seriously affects the appearance of the object.
> >
> > **This preliminary data validates the importance of these systematic failure patterns.** We will formalize this into a comprehensive quantitative analysis in the camera-ready version, providing a clear roadmap for future improvements in the field. We are grateful for this suggestion, as it will undoubtedly strengthen the impact of our work.

---

> > ### Author Response · Authors · 2025-08-08
> > **Thank you for your positive feedback!**
> >
> > Dear Reviewer VN3G,
> >
> > We sincerely thank the reviewer for their detailed feedback and for acknowledging the potential of our proposed improvements.
> >
> > We wanted to briefly highlight that we have added **three responses in the public discussion thread** to address the issues you left behind. **(Stratified results per superclass and robustness of long-tail distributions ! The visualizations showing how reference anchors refine across decoding stages ! Deeper failure analyses !)** We believe these materials substantially strengthen our claims and hope they will provide the necessary evidence to reconsider the score.
> >
> > We would like to resolve any and all remaining points. Our goal is to ensure the final manuscript has your full support and confidence. We would be most grateful if you could consider improving your rating.
> >
> > Thank you again for your constructive feedback and the time and effort you dedicated to reviewing our work!
> >
> > Thank you once more for your support, valuable time and consideration.

---

### Official Review · Reviewer_P9vU · 2025-06-30

**Clarity:** 2
**Significance:** 2
**Originality:** 2
**Rating:** 4
**Confidence:** 4

**Summary:**

The paper proposes a new task, Spatial Aerial Video Grounding. UAV-SVG, a large-scale benchmark. They further introduce SAVG-DETR, a transformer-based architecture designed to address the challenges of aerial scenes, including small object size, camera motion, and complex spatial context.

**Questions:**

See the weakness

**Ethical Concerns:**

["Major Concern: Data privacy, copyright, and consent", "Major Concern: Data quality and representativeness"]

**Final Justification:**

The authors addressed my concerns

**Limitations:**

No. While the paper discusses some technical limitations, it overlooks potential negative societal impacts.

**Quality:**

2

**Strengths And Weaknesses:**

Strengths

- The paper is logically structured, and most technical sections are clearly explained despite the method's complexity.

- The paper presents a well-engineered framework (SAVG-DETR). The evaluations include comparisons with baselines, thorough ablations, and qualitative visualizations.

- The introduction of the UAV-SVG large-scale benchmark, which is underrepresented in existing spatial video grounding datasets.

- The proposed SAVG-DETR introduces several original architectural elements that directly address the challenges of aerial video grounding.

Weaknesses

- While the overall organization is sound, certain sections, particularly the encoder/decoder descriptions, are overly dense and could be more readable. Long technical paragraphs, repeated use of phrases like "computational explosion."

- The benchmark relies partially on LLM-generated expressions (Gemini). While this enables scale, the paper lacks a deeper discussion or analysis of potential risks.
- The evaluation is heavily focused on UAV-SVG. It remains unclear how well SAVG-DETR generalizes to other grounding datasets or aerial sources (e.g., satellite or oblique-view videos).

- Although the authors decouple the encoder to reduce complexity, the final model still requires substantial computational resources (e.g., 203.08 GFLOPs and 28.7 GB of memory, as shown in Table 4).

---

> ### Author Rebuttal · Authors · 2025-07-30
>
> >**Q1**: While the overall organization is sound, certain sections, particularly the encoder/decoder descriptions, are overly dense and could be more readable. Long technical paragraphs, repeated use of phrases like "computational explosion."
>
> **A1**: Thank you for this constructive feedback on the paper's readability. We agree that clarity is crucial, especially in the technical sections. The density of the methodology section was partly a consequence of fitting our detailed architecture into the page limit.
>
> To address this, we will thoroughly revise Sections 4.1 and 4.2 in the camera-ready version to improve their structure and flow. Our planned revisions include:
> 1. Breaking Down Paragraphs: We will divide the long, dense paragraphs into shorter, more digestible ones, with each focusing on a single component or concept.
> 2. Using Structural Elements: We will incorporate bulleted or numbered lists to delineate the key steps within our proposed encoder and decoder modules. This will make the architecture's logic easier to follow.
> 3. Refining Language: We will carefully revise the text to improve the prose and vary our phrasing, avoiding the overuse of terms like "computational explosion".
>
> We are confident that these revisions will make the core methodology of our paper significantly more accessible to readers.
>
> >**Q2**: The benchmark relies partially on LLM-generated expressions (Gemini). While this enables scale, the paper lacks a deeper discussion or analysis of potential risks.
>
> **A2**: Thank you for raising this important point about the responsible use of  LLMs in dataset creation. We recognize the potential risks, such as factual inaccuracies or biases, and we designed a rigorous, human-centric pipeline specifically to mitigate them.
> Our defense against these risks is a three-step process, as detailed in Section B.1 of the supplementary material :
> 1. Guided Generation with Human Priors: We did not use the LLM in a zero-shot fashion. Instead, we provided Gemini with "object-specific prompt with human priors", which included manually annotated object descriptions from the original WebUAV-3M dataset. This step grounds the LLM's output in factual, human-verified information from the start.
> 2. LLM-based Augmentation: The model was then used to augment these descriptions, generating more diverse and descriptive referring expressions.
> 3. Rigorous Manual Verification: This is the most critical step. Every single expression generated by the LLM was subjected to meticulous manual checking and correction by our annotation team. As we state, "Each expression is manually checked to determine whether the described attributes are correct and whether the referred object can be uniquely distinguished. Correct any errors that exist... If an instance is difficult to describe uniquely and precisely... discard this sample". This extensive verification process took approximately three months to complete.
>
> To make this clearer in the paper, we will add a dedicated discussion in the camera-ready version that explicitly addresses the potential risks of LLM usage and details how our human-in-the-loop pipeline systematically neutralizes them.
>
> >**Q3**: The evaluation is heavily focused on UAV-SVG. It remains unclear how well SAVG-DETR generalizes to other grounding datasets or aerial sources (e.g., satellite or oblique-view videos).
>
> **A3**: We appreciate this question regarding the generalizability of our model. Our main objective in this work was to introduce and formally define the novel Spatial Aerial Video Grounding (SAVG) task , a challenging and underexplored area within the low-altitude domain. To this end, we constructed the first large-scale benchmark, UAV-SVG, which presents a unique set of challenges not found in existing datasets. These include:
> * A vast number of extremely small objects, often only a few pixels in size.
> * Rapid and complex camera motion inherent to UAV platforms.
> * A bird's-eye view perspective encompassing wide, dense, and complex scenes.
>
> We agree that exploring the adaptability of our SAVG-DETR is a valuable avenue for future research.
> * Other Grounding Datasets: While we believe our model could be adapted, its core strengths are in handling the extreme conditions of UAV data, which are less prevalent in ground-level datasets.
> * Other Aerial Sources: While satellite video is very important, there is a lack of SVG datasets for satellite video. Applying our method there would be a compelling, but different, research problem.
>
> Our proposed model, SAVG-DETR, was specifically engineered to tackle these distinct challenges. Key components, such as the decoupled multi-scale encoder , the multi-level progressive decoder for enhancing small object grounding , and the offset-based decoding paradigm for improving prediction consistency across frames with erratic motion, were designed with the explicit properties of the SAVG task in mind.
>
> Our evaluation is intentionally focused on the UAV-SVG benchmark, as it is the most appropriate and direct way to validate the effectiveness of our proposed solutions against the specific problems we set out to solve. As our experiments in Table 2 demonstrate, state-of-the-art methods designed for standard ground-level video grounding show a significant performance drop on UAV-SVG, which confirms that SAVG is a distinct and more challenging task requiring a specialized approach.
>
> Therefore, our main goal was to demonstrate that SAVG-DETR sets a new SOTA on the challenging task it was designed for. While we believe the architectural principles are sound, evaluating its direct generalization to other domains is a significant research direction in itself and falls outside the scope of this foundational work. We will state this more clearly in our "Limitations and Future Work" section in the camera-ready paper, positioning the exploration of our model's applicability to other datasets as an important avenue for future investigation.
>
> Thank you again for your constructive feedback. We hope this clarifies that our focused evaluation was a deliberate choice to rigorously assess our solution against the novel problem we introduced.
>
>
> >**Q4**: Although the authors decouple the encoder to reduce complexity, the final model still requires substantial computational resources (e.g., 203.08 GFLOPs and 28.7 GB of memory, as shown in Table 4).
>
> **A4**: Thank you for this astute observation. We acknowledge that our model is computationally intensive in absolute terms. Our claim of reducing complexity needs to be understood in the context of our specific architectural goals.
>
> The core challenge we address is incorporating multi-scale features into a multi-modal transformer architecture, which is essential for grounding small objects in high-resolution aerial videos. A naive implementation of this (Variant C in Table 4) is computationally infeasible and resulted in an out-of-memory error on our hardware. Our key innovation—decoupling the encoder into an intra-scale multi-modal branch and a cross-scale visual-only branch—is what makes this powerful approach computationally tractable. As shown in Table 4, our decoupling strategy not only works but is also more efficient than a more straightforward cascaded design (Variant D), reducing FLOPs by 17% while improving performance.
>
> The high resource usage is a direct trade-off for the significant leap in performance on the challenging UAV-SVG benchmark. The extensive multi-scale processing is precisely what allows SAVG-DETR to outperform other methods by a large margin (e.g., +7.02% m_vIoU over the efficient SGFDN model ).
>
> We agree that efficiency is critical for practical applications. As stated in our "Limitations and Future Work" section, exploring "smaller sizes and faster models"  is a key priority. We will emphasize this trade-off and the need for future work on model efficiency more explicitly in the final version of the paper.

---

> > ### Author Response · Authors · 2025-08-04
> >
> > Dear Reviewer P9vU,
> >
> > Thank you again for your thoughtful and constructive comments!
> >
> > As the discussion phase is drawing to a close, we would like to kindly ask whether our responses have adequately addressed your concerns. If there are any clarifications needed, we would be pleased to provide further details.
> > We would like to resolve any and all remaining points. Our goal is to ensure the final manuscript has your full support and confidence. We would be most grateful if you could consider improving your rating.
> >
> > Best regards,
> >
> > Authors of Submission #546

---

> > ### Comment · Reviewer_P9vU · 2025-08-06
> >
> > Thank you for the detailed rebuttal. While I appreciate the clarifications, key concerns remain insufficiently addressed. In particular, the lack of evaluation beyond UAV-SVG limits the model's demonstrated generalizability. The reliance on LLM-generated annotations still lacks rigorous analysis of potential biases or quality metrics. Promised clarity and efficiency improvements are deferred to the camera-ready version, making it difficult to assess their effectiveness during review. I maintain my original assessment.

---

> > > ### Author Response · Authors · 2025-08-07
> > > **Evaluation of model generalizability beyond UAV-SVG !**
> > >
> > > >**Q1:** In particular, the lack of evaluation beyond UAV-SVG limits the model's demonstrated generalizability.
> > >
> > > **A1**: We sincerely thank the reviewer for their insightful feedback and acknowledge the importance of demonstrating our model's generalizability. We agree that evaluation on other grounding benchmarks is crucial for a comprehensive assessment.
> > >
> > > **After obtaining the reviewers' valuable suggestions in the early stage, we began the generalization experiment of SAVG-DETR.** Since there is indeed a lack of other aerial datasets for this task, we are currently conducting experiments on two widely-used natural scene datasets: HCSTVG-v1 and VidSTG (declarative sentences). We present the complete results below, comparing our method with recent state-of-the-art (SoTA) models.
> > >
> > >
> > > |Methods|m_vIoU|vIoU@0.3|vIoU@0.5|
> > > |-|:------------:|:------------:|:------------:|
> > > |TubeDETR[CVPR22]|32.40|49.80|23.50|
> > > |STCAT[NeurIPS22]|35.09|57.67|30.09|
> > > |STVGFormer[CVPR23]|36.90|62.20|34.80|
> > > |VideoGrounding-DINO[CVPR24]|38.25|62.47|36.14|
> > > |CG-STVG[CVPR24]|38.40|61.50|36.30|
> > > |**SAVG-DETR**|**37.22**|**60.63**|**35.45**|
> > > Table 1. Comparison with existing SoTA methods on HCSTVG-v1 test set(%).
> > >
> > >
> > > |Methods|m_vIoU|vIoU@0.3|vIoU@0.5|
> > > |-|:------------:|:------------:|:------------:|
> > > |TubeDETR[CVPR22]|30.40|42.50|28.20|
> > > |STCAT[NeurIPS22]|33.14|46.20|32.58|
> > > |STVGFormer[CVPR23]|33.70|47.20|32.80|
> > > |VideoGrounding-DINO[CVPR24]|34.67|48.11|33.96|
> > > |CG-STVG[CVPR24]|34.00|47.70|33.10|
> > > |**SAVG-DETR**|**33.89**|**47.05**|**32.95**|
> > > Table 2. Comparison with existing SoTA methods on VidSTG (declarative sentences) test set(%).
> > >
> > >
> > > **Analysis of Results:**
> > >
> > > 1. As the results indicate, while our SAVG-DETR does not set a new SoTA on these natural scene datasets, it demonstrates **highly competitive performance, placing it on par with or very close to the leading methods**.
> > >
> > > 2. This outcome is aligned with our central motivation. The primary architectural innovations of SAVG-DETR, particularly the introduction of multi-scale visual features and the progressive decoder, were **specifically designed to tackle the unique challenges present in aerial videos—most notably, the extreme prevalence of small objects.** As we detailed in our supplementary material (Figure 7), the scale of target objects in our UAV-SVG benchmark is significantly smaller than in datasets like HCSTVG and VidSTG.
> > >
> > > 3. In standard grounding datasets where objects are typically larger and more distinct, **the benefits of our multi-scale approach are less pronounced, yet the model remains robust.** Conversely, on our UAV-SVG benchmark, these components are critical and enable SAVG-DETR to achieve its outstanding performance.
> > >
> > > Therefore, the fact that our model achieves such competitive results on standard benchmarks **validates its robust and generalizable foundation.** Its superior performance on UAV-SVG further highlights its targeted effectiveness in solving the challenging and previously underexplored problem of spatial aerial video grounding.
> > >
> > > We are confident that these new results effectively prove the generalizability of our model's architecture. **We will, of course, add these tables and the accompanying analysis to the final version of our paper** to thoroughly address this important point. Thank you again for pushing us to strengthen our work.

---

> > > ### Author Response · Authors · 2025-08-07
> > > **Rigorous analysis of potential biases and quality metrics !**
> > >
> > > >**Q2:** The reliance on LLM-generated annotations still lacks rigorous analysis of potential biases or quality metrics.
> > >
> > > **A2:** We thank the reviewer for this critical question and appreciate the opportunity to elaborate on the rigor of our data annotation process. We understand the concern regarding potential biases in LLM-generated content and agree that a detailed quality analysis is essential.
> > >
> > > 1. Potential Biases
> > >
> > > - We wish to clarify that our data construction is a human-centric pipeline, where the LLM serves as a powerful tool to assist human annotators, rather than a fully autonomous generator. The entire process was designed with human oversight at its core to ensure high-quality, non-ambiguous annotations.
> > >
> > > - To mitigate potential LLM biases from the outset, the prompts provided to the model already included human-prior descriptions of the target object. This grounded the LLM's generation process in manually verified information, significantly reducing the likelihood of unconstrained or biased outputs.
> > >
> > >
> > > To provide the rigorous analysis requested, we have reorganized the log data of the manual verification process and provide our verification protocol and quality metrics as follows:
> > >
> > > 2. Data Verification Protocol
> > >
> > > Our dataset consists of 3,564 videos and 17,820 language annotations. The verification involved a two-stage human review:
> > >
> > > - Initial Check: Two annotators served as initial checkers. Their task was to assess each generated expression to determine if the described attributes (e.g., color, shape, size, spatial position, quantity, action state (motion state), relationship) were correct and if the expression uniquely identified the referred object in the video. If any issues were found, they were manually corrected.
> > >
> > > - Final Check: A third, senior annotator acted as the final checker. They reviewed all expressions. If any ambiguities or errors persisted, the sample was returned to the initial checkers for another round of revision.
> > >
> > > **3. Detailed Quality Metrics**
> > >
> > > This human-in-the-loop process allowed us to quantify the quality and correction process:
> > >
> > > - Initial Correction Rate: **The correction rate by our initial checkers was 13%.** This demonstrates that the manual review was non-trivial and essential for refining the raw LLM outputs.
> > >
> > > - Error Analysis: We found that errors were most common in complex, dynamic attributes that require spatio-temporal reasoning. These included spatial position, motion state, and object relationships. For example, an initial expression might state, "The black all-terrain vehicle that is driving on the snowy surface, going from top to bottom in the video." This was flagged as an error because the drone's own movement created this illusion; a human checker corrected it to reflect the vehicle's actual trajectory from bottom to top.
> > >
> > > - Final Quality Assurance: **The rejection rate by the final checker was a mere 0.11%.** This extremely low rate for the final pass demonstrates the effectiveness of our two-stage protocol in achieving our goal of non-ambiguous and high-quality annotations.
> > >
> > > To further substantiate these points, **we will add a detailed description of this verification protocol, the quality metrics, and several illustrative examples of corrected annotations to the final version of our paper.** We believe this detailed breakdown demonstrates the meticulous care taken in constructing the UAV-SVG benchmark and thoroughly addresses the concerns about the quality and potential biases of our annotation process.

---

> > > ### Author Response · Authors · 2025-08-07
> > > **Complexity and computational resources !**
> > >
> > > **A3**: We sincerely thank the reviewer for this insightful comment and appreciate the opportunity to clarify the fairness and rationale behind our computation comparisons. We respectfully argue that the comparison is indeed appropriate and highlights the core contributions of our work.
> > >
> > > 1. To effectively address the task of aerial object localization, we found it necessary to introduce multi-scale visual features. This makes a fair comparison with single-scale methods like TubeDETR or SGFDN challenging. SGFDN is a method specifically designed for efficiency, and while it has high computational efficiency, its performance is very low. TubeDETR is more efficient than our method, but its performance is also lower.
> > >
> > > 2. Directly incorporating multi-scale visual features into TubeDETR/STCAT leads to a computational explosion. The analysis of Variant C in Table 4 proves this point; this direct multi-scale, multi-modal transformer approach resulted in prohibitive memory requirements and a substantial increase in FLOPs. Our analysis of variants A through E demonstrates that our proposed method achieves the best balance between performance and efficiency among the feasible multi-scale approaches.
> > >
> > > 3. This situation also explains why a significant body of research is dedicated to designing multi-scale DETR methods. Standard DETR-based models do not perform well on UAV aerial targets with their unique challenges. Introducing multi-scale features inevitably increases complexity, which is an unavoidable trade-off for achieving higher accuracy in this domain. Therefore, we believe our computation experiments are reasonable, highlighting the effectiveness of our proposed architecture in handling these complexities.
> > >
> > >
> > > 4. The core challenge we address is incorporating multi-scale features into a multi-modal transformer architecture, which is essential for grounding small objects in high-resolution aerial videos. A naive implementation of this (Variant C in Table 4) is computationally infeasible and resulted in an out-of-memory error on our hardware. Our key innovation—decoupling the encoder into an intra-scale multi-modal branch and a cross-scale visual-only branch—is what makes this powerful approach computationally tractable. As shown in Table 4, our decoupling strategy not only works but is also more efficient than a more straightforward cascaded design (Variant D), reducing FLOPs by 17% while improving performance.

---

> > > ### Author Response · Authors · 2025-08-07
> > > **Thank you for your positive feedback！**
> > >
> > > Dear Reviewer P9vU,
> > >
> > > Thank you again for your positive engagement. We are pleased that our response addressed your some concerns!
> > >
> > > We wanted to briefly highlight that we have added **three responses in the public discussion thread** to address the issues you left behind. (**Evaluation of model generalizability beyond UAV-SVG ! Rigorous analysis of potential biases and quality metrics ! Complexity and computational resources !**) We believe these results provide significant additional evidence for our method's effectiveness, and we are happy to answer any questions you may have about them as well.
> > >
> > > We would like to resolve any and all remaining points. Our goal is to ensure the final manuscript has your full support and confidence. We would be most grateful if you could consider improving your rating.
> > >
> > > Thank you again for your constructive feedback and the time and effort you dedicated to reviewing our work!
> > >
> > > Thank you once more for your support, valuable time and consideration.

---

> ### Comment · Area_Chair_1B1c · 2025-08-05
>
> Hi P9vU,
>
> Please check the author's feedback, evaluate how it addresses the concerns you raised, and post any follow-up questions to engage the authors in a discussion. Please do this ASAP.
>
> Thanks.

---

### Official Review · Reviewer_kDhD · 2025-07-03

**Clarity:** 3
**Significance:** 3
**Originality:** 3
**Rating:** 5
**Confidence:** 4

**Summary:**

This paper makes contributions to UAV-based spatial video grounding (SVG) by introducing UAV-SVG, a large-scale benchmark addressing the lack of specialized datasets for aerial video analysis, and proposing SAVG-DETR, a novel framework with task-specific designs that outperforms existing methods.

**Questions:**

Please see the Weaknesses part above.

**Ethical Concerns:**

["NO or VERY MINOR ethics concerns only"]

**Final Justification:**

Thank you for addressing my previous concerns. I maintain my original score.

**Limitations:**

yes

**Paper Formatting Concerns:**

I do not notice any major formatting issues in this paper.

**Quality:**

4

**Strengths And Weaknesses:**

Strengths:
1) The paper makes a valuable contribution by establishing UAV-SVG, a novel and challenging benchmark for spatial video grounding in aerial scenes, addressing a critical gap in the field.
2) The proposed SAVG-DETR architecture demonstrates thoughtful design choices tailored to the unique demands of aerial video analysis, leveraging transformer-based approaches effectively.
3) The method achieves state-of-the-art performance, supported by comprehensive experiments and rigorous evaluation.
4) The manuscript is well-written, with clear explanations and thorough analysis, including insightful discussions of failure cases that enhance future research directions.

Weaknesses:
1) Figure 3(b) requires revision, as the origin and purpose of the input blue and pink boxes are unclear. These elements should be explicitly explained or visually adjusted to align with the output from phase (a) to avoid confusion.
2) The problem definition, currently relegated to supplementary materials, should be integrated into the main Methodology section to improve readability and ensure immediate context for readers.
3) I believe this aerial SVG task is rather difficult when it comes to grounding upon dense objects environment (a common scenario in aerial imagery). The authors should provide additional analysis in the camera-ready version to clarify whether performance limitations stem from task complexity or methodological constraints, offering potential pathways for improvement.

---

> ### Author Rebuttal · Authors · 2025-07-30
>
> >**Q1**: Figure 3(b) requires revision, as the origin and purpose of the input blue and pink boxes are unclear. These elements should be explicitly explained or visually adjusted to align with the output from phase (a) to avoid confusion.
>
> **A1**: Thank you for your insightful feedback and for highlighting this point of confusion. We agree that the clarity of Figure 3(b) can be significantly improved. The blue and pink boxes are indeed critical inputs to our proposed encoder, and we appreciate the opportunity to clarify their roles and origins.
> 1. The Blue Boxes (Visual Features): The blue boxes represent the high-level visual features $S_5$ extracted by the Visual Encoder in Figure 3(a). All three scales of visual features ($S_3$, $S_4$, $S_5$) represent by the blue boxes of varying shades.
> 2. The pink box represents a learnable frame token ($F^f$). As we explain in Section 4.1 and illustrate in detail in Figure 13 of the supplementary material, this token is introduced within the Intra-scale Multi-modality Interaction Branch. Its purpose is to act as an aggregator, capturing the most relevant spatial context of the referred object for each frame after the vision and language features have interacted.
>
> To address this, we will revise Figure 3 in the camera-ready version to make the data flow more explicit and intuitive.
>
>
> >**Q2**: The problem definition, currently relegated to supplementary materials, should be integrated into the main Methodology section to improve readability and ensure immediate context for readers.
>
> **A2**: We appreciate this excellent suggestion. The reviewer is correct that placing the problem definition at the beginning of the Methodology section is standard practice and enhances the paper's flow and readability. Our initial decision to place it in the supplementary material (as noted in Section 4, line 163 ) was solely due to the strict space constraints of the initial submission.
>
> In the camera-ready version, we will move the "Problem Definition" from Section C.1 of the supplementary material  to the beginning of Section 4 (Methodology). This will provide readers with immediate formal context for the SAVG task before we describe the technical details of our SAVG-DETR framework, improving the overall structure and coherence of the paper.
>
> >**Q3**: I believe this aerial SVG task is rather difficult when it comes to grounding upon dense objects environment (a common scenario in aerial imagery). The authors should provide additional analysis in the camera-ready version to clarify whether performance limitations stem from task complexity or methodological constraints, offering potential pathways for improvement.
>
> **A3**: Thank you for this insightful comment. We agree that grounding in dense object environments is a core challenge of the SAVG task and a primary motivation for our work. As noted in our introduction, aerial videos often feature a "wide field of view and a large scene scale, containing dense and numerous objects". This inherent task complexity, combined with challenges like small object scale and background clutter, pushes the boundaries of current grounding methods.
>
> The performance limitations observed are a result of both the profound difficulty of the task and the constraints of current methodologies.
> 1. Task Complexity: The UAV-SVG benchmark was intentionally constructed to include such challenging scenarios (e.g., Expression 2 and 5 in Figure 5 , showing dense vehicles). These scenarios require not only recognizing an object but also understanding its complex spatial relationships with many similar-looking distractors, often described by subtle linguistic cues.
> 2. Methodological Advances and Constraints: Our proposed SAVG-DETR is specifically designed to address this. The multi-modality multi-scale spatio-temporal encoder and the hierarchical progressive spatial decoder  work in concert to integrate high-level semantic context with low-level spatial details, which is crucial for distinguishing a target from nearby distractors. The superior performance of our method over strong baselines in Table 2  demonstrates that our approach is a significant step forward. However, we acknowledge that our method still has constraints. When objects are extremely similar and densely packed, precisely interpreting complex language (e.g., "the second actor counting from left to right" ) remains an open research problem.
>
> In the camera-ready version, we will add a dedicated subsection for a more detailed analysis of this issue. This new section will:
> 1. Analyze performance in dense scenes, drawing on qualitative examples from our failure analysis (e.g., Figure 17 ) to discuss specific cases where our model succeeds or fails in distinguishing targets from dense distractors.
> 2. Clarify the primary sources of error, attributing them to either highly ambiguous visual information or the model's limitations in processing complex relational language.
> 3. Offer more detailed pathways for improvement, expanding on our Limitations section . We will suggest future research directions, such as incorporating explicit object-to-object relation modeling modules or leveraging more advanced large language models for deeper semantic reasoning to better handle these complex, dense scenarios.

---

> ### Comment · Area_Chair_1B1c · 2025-08-05
>
> Hi kDhD ,
>
> Please check the author's feedback, evaluate how it addresses the concerns you raised, and post any follow-up questions to engage the authors in a discussion. Please do this ASAP.
>
> Thanks.

---

### Author Response · Authors · 2025-08-08
**Clarifying the Contributions & Summarizing the Rebuttal !**

We thank all the valuable feedback and comments from reviewers that have helped to improve our paper! We also thank the reviewers for appreciating the intuitive and thoughtful design of the SAVG-DETR (Reviewer kDhD, VN3G, P9vU), the effectiveness of SAVG-DETR (Reviewer VN3G, GFfV), comprehensiveness of the rigorous evaluation (Reviewer kDhD, P9vU), as well as great presentation and writing (Reviewer kDhD, P9vU, VN3G). We summarized the contributions of this paper and addressed the main concerns of the reviewers.

## Contributions

**------------------- Task Contributions ------------------**

1) **A groundbreaking task:** The paper introduces a novel task called Spatial Aerial Video Grounding (SAVG), which aims to localize objects in UAV videos based on natural language queries.

2) **Practical application significance:** We can use UAVs to localize specific objects in various scenarios from the sky view and obtain a much more holistic grounding. As the low-altitude economy takes off, many tasks currently need to be performed in the sky, such as UAV-based goods delivery, traffic/security patrol, and scenery tours.

3) The current spatial video grounding primarily focuses on the fixed or handheld camera perspectives in simple scenes. SAVG introduced **an aerial platform moving in the sky, a more complex scene.**

**------------------- Benchmark Contributions ------------------**

1) The introduction of UAV-SVG **addresses a critical gap in the field**, providing a new **large-scale benchmark** that reflects real-world UAV challenges like small objects, illumination variations, and fast motion. The dataset's scale (2M+ frames) and **diversity (216 categories)** make it a valuable resource for future research, which is underrepresented in existing datasets.

2) UAV-SVG introduces a **more challenging dataset** for the video grounding task and will be a valuable resource for the community. The subsequent video grounding work will need to be verified on UAV-SVG, as it has **12 unique characteristics**, challenges, and a richer variety of object categories.

3) A comprehensive evaluation and analysis were conducted on the existing SoTA methods.

**------------------- Technical Contributions ------------------**

The proposed SAVG-DETR architecture demonstrates thoughtful design choices tailored to the unique demands of aerial video analysis, a transformer-based architecture with three key innovations:

1) **Decoupled encoder:** Separates multi-modality and multi-scale spatio-temporal modeling to reduce computational complexity.

2) **Hierarchical progressive decoder:** Enhances small object grounding by integrating multi-scale features into language modulation and progressively refining predictions.

3) **Offset-based decoding:** Improves prediction consistency across frames by iteratively updating reference anchors.


## Main concerns addressed

**1. Bias in Benchmark:** We elaborated on our detailed data construction methodology, which is inherently designed to mitigate biases to the greatest extent. We provided a bias analysis and detailed quality metrics of our manual verification process.

**2. Generalizability of SAVG-DETR:** We provided the performance results and a detailed analysis of SAVG-DETR on natural videos.

**3. Computational Efficiency:** We clarified the fairness of our computational comparisons. Our method achieves a balance between performance and efficiency while incorporating multi-scale visual features.

**4. Robustness of SAVG-DETR on Long-Tailed Distributions:** We provided performance results for SAVG-DETR and other methods on each superclass and analyzed the impact of the long-tailed distribution on performance.

**5. Failure Analysis:** We conducted a quantitative investigation of failure cases, providing a more detailed systematic analysis. The failure rates under patterns such as objects below a certain size, fast motion, and viewpoint changes can guide future research.

---

### Note · Authors · 2025-08-12

We would like to express our sincere gratitude to the AC and SAC. We also thank the reviewers for appreciating the intuitive and thoughtful design of the SAVG-DETR, the effectiveness of SAVG-DETR, comprehensiveness of the rigorous evaluation, as well as great presentation and writing.

**1) Task Contribution:** All reviewers have acknowledged the importance and contribution of the proposed task.

**2) Benchmark Contribution:**

- Reviewer GFfV inquired about data details and potential bias, but has already acknowledged our clarification.

- Reviewer P9vU inquired about potential risks of the data generation. We have provided an in-depth discussion on bias and detailed quality metrics. We are confident this can address the reviewer's concerns.

- All points raised by the Ethics Reviewer are addressable. We have provided our responses and have committed to implementing the improvements in the final version of the paper.

**3) Technical Contribution:** After we resolved a slight misunderstanding from Reviewer GFfV regarding SAVG-DETR, all reviewers have now acknowledged our technical contributions.

**4) Experimental Section:** Concerns Addressed

1. Reviewer P9vU:

- Generalizing SAVG-DETR to other grounding datasets: We have provided detailed experiments that resolve this concern.

- High computational cost: We have provided a detailed explanation. SAVG-DETR is a multi-scale method. Direct comparison with existing methods is not entirely fair. When multi-scale features are introduced, we achieve the best balance between performance and efficiency.

2. Reviewer GFfV: Unfair computational comparisons: Following our response, the reviewer has acknowledged our clarification.

3. Reviewer VN3G:

- Discussion on long-tailed distribution and robustness: We provided results for SAVG-DETR and other methods on each superclass and analyzed the impact of the long-tailed distribution.

- Visualization of the decoding stages: We have provided a detailed response and have committed to including the visualization in the final version.

- Systematic failure case analysis: As per the reviewer's suggestion, we have provided a quantitative investigation and a more detailed systematic analysis of failure cases during the discussion stage.

The paper's contributions span the areas of task, benchmark, and methodology. These have been summarized in detail in the "Official Comment: Clarifying the Contributions & Summarizing the Rebuttal!" (**Please look at the chat box below**).

---

### Decision · Program_Chairs · 2025-09-17

**Decision:**

Accept (poster)

**Comment:**

Paper Summary:
This paper introduces a new large-scale benchmark for Spatial Video Grounding (SVG) specifically tailored to UAV footage. Concurrently, the authors propose SAVG-DETR, a Transformer-based framework designed to handle the unique challenges of aerial scenes, such as small objects and complex camera motion.

Main Strengths and Weaknesses:
As acknowledged by all reviewers, the primary strength of the paper is the creation of the challenging UAV-SVG benchmark, which addresses a critical gap in the field and provides a valuable resource for future research. A key weakness, noted by multiple reviewers, is the high computational cost (in terms of GFLOPs and memory) of the proposed SAVG-DETR model, which raises questions about its practical applicability on resource-constrained UAV hardware.

Rebuttal Analysis:
The rebuttal effectively addresses the concerns raised by the reviewers. Reviewer VN3G was initially unconvinced but was satisfied after a second round of responses that included additional experiments, ultimately concluding the paper meets publication standards. Reviewer GFfV found their questions answered and slightly raised their score, though they remained somewhat skeptical about the significance of the contribution of the dataset compared to existing work.

Final Justification and Recommendation:
All four reviewers converged on a positive recommendation, viewing the paper as a solid technical contribution. The core contributions—a much-needed dataset for aerial video analysis and a corresponding high-performing model—are significant. The authors successfully addressed most of the technical and clarity issues raised during the review process. While valid concerns remain regarding the computational efficiency for real-world deployment, the consensus is that the strengths and robust contributions to an under-explored area outweigh these weaknesses.